# A multi-encoder variational autoencoder controls multiple transformational features in single-cell image analysis

Luke Ternes[1], Mark Dane [1], Sean Gross[1], Marilyne Labrie [2], Gordon Mills[2], Joe Gray[1], Laura Heiser [1✉] & Young Hwan Chang [1✉]

Image-based cell phenotyping relies on quantitative measurements as encoded representations of cells; however, defining suitable representations that capture complex imaging features is challenged by the lack of robust methods to segment cells, identify subcellular compartments, and extract relevant features. Variational autoencoder (VAE) approaches produce encouraging results by mapping an image to a representative descriptor, and outperform classical hand-crafted features for morphology, intensity, and texture at differentiating data. Although VAEs show promising results for capturing morphological and organizational features in tissue, single cell image analyses based on VAEs often fail to identify biologically informative features due to uninformative technical variation. Here we propose a multi-encoder VAE (ME-VAE) in single cell image analysis using transformed images as a self-supervised signal to extract transform-invariant biologically meaningful features, including emergent features not obvious from prior knowledge. We show that the proposed architecture improves analysis by making distinct cell populations more separable compared to traditional and recent extensions of VAE architectures and intensity measurements by enhancing phenotypic differences between cells and by improving correlations to other analytic modalities. Better feature extraction and image analysis methods enabled by the ME-VAE will advance our understanding of complex cell biology and enable discoveries previously hidden behind image complexity ultimately improving medical outcomes and drug discovery.

[1] Biomedical Engineering Department, Oregon Health & Science University, Portland, OR 97239, USA. [2] Cell, Developmental and Cancer Biology Department, Oregon Health & Science University, Portland, OR 97239, USA. ✉email: heiserl@ohsu.edu; chanyo@ohsu.edu

Understanding cellular changes and phenotypic pathways at the single-cell level is becoming increasingly important because it creates a comprehensive understanding of cell state and cell-to-cell heterogeneity. Multiple analytical tools are available to extract, normalize, and evaluate single-cell RNA sequencing (scRNAseq) data[1–3]. Until recently, analyzing single-cell imaging data in a similar fashion was limited to extracting mean intensity profiles, predefined shape, textural, and morphological features, and images stained with only a few markers. Emerging multiplexed imaging technologies such as cyclic immunofluorescence (CYCIF)[4], multiplexed immunohistochemistry[5], CO-Detection by indEXing (CODEX)[6], and Multiplexed Ion Beam Imaging[7] create images comprised of a large number of markers, expanding the depth of information. Robust analytical methods for high-dimensional multiplexed imaging data, however, are still needed. One limitation with analyzing highly multiplexed single-cell images is the ability to extract biologically meaningful information on staining localization patterns that indicate divergent cell states. Single-cell imaging data has morpho-spatial information not captured using simple mean intensity information, with successful quantification of these features potentially leading to improved analysis and understanding[8].

The classical approach for image feature extraction is manually creating a list of desired features and predefined metrics to quantify them. This is biased toward known and easily measured features and can miss subtle but important features. More robust image feature extraction has been employed using deep learning architectures such as the Variational Autoencoder (VAE)[9] in other domains where feature representation can be automatically generated without supervising information or prior knowledge. However, the problem with VAE feature extraction in single-cell imaging is that there are typically unimportant or uninformative features driving differences between biologically similar images and skewing the results in undesired ways[10]. In single-cell imaging data, these unimportant features include any form of basic image transformation such as rotation, transposition, affine/skew, and stretching. Despite having the same underlying information, the common uninformative features in transformed images distract deep learning architectures so that they ignore most of the biologically relevant features[11–15]. This holds true in single-cell images, where VAEs frequently ignore biologically meaningful features and focus on recreating the transformational features which have a high variance across the dataset. When these features are known and controllable transformations, they can be used for a self-supervised signal to extract invariant features with respect to a set of transformations during model training. Untailored deep learning architectures are unable to overcome these uninformative features unless some modification is made to either their architecture or objective functions[11–15]. Many recent works propose changing autoencoder architectures to coupled networks or using multiple latent dimensions to overcome this without the need for biased hyperparameter tuning and data normalization[10,16–18]. Similar methodologies have also been explored that seek to correct transformative features with coupled networks, direct latent space modifications, novel layer architectures, and training networks with combinations of corrected and uncorrected image data[11–13,15,19,20]. Most of these corrected architectures, however, only target one specified feature and cannot generalize to other features without further modification. Two examples of recent architectures that use modifications to the objective function are the β-VAE[21] and the invariant C-VAE[22], which attempt to apply pressure to the model such that it will prioritize a more regularized encoding space and be more interpretable and invariable to specific features.

Here we propose a method for single-cell image feature extraction that removes specified uninformative features by making them uniform and invariant across the reconstructions, using modified pairs of transformed input and output images by self-supervised transformation, and utilizing multiple encoding blocks. Using this multi-encoder VAE (ME-VAE) to control for multiple transformational features, we highlight its ability to extract biologically meaningful and transform-invariant single-cell information and better separate biologically distinct cell populations.

## Results

**Controlling for uninformative features**. When a transformational feature varies across a single-cell imaging dataset, standard VAEs extract only the dominant component to reconstruction. When rotation varies from image to image, reconstructions along the principal component walk[23] only constitute the angle of the cell, and downstream analysis is heavily skewed by this extracted component (Fig. 1a). In another dataset where polar orientation is the dominant feature, we observe the same behavior (Fig. 1b); VAEs only extract the dominant uninformative features, ignoring subtle but informative features necessary for detailed reconstruction.

In order to overcome model hypersensitivity to dominant uninformative features, several architectures were proposed and tested to learn the latent space while attempting to ignore uninformative features (Fig. 1c–g). A standard VAE without control for uninformative features was used as a baseline and shows a high correlation between the embedded components and the respective feature metrics (Fig. 1c and Supplementary Fig. 1a). When a single factor is controlled (e.g., rotation), it becomes uncorrelated to all VAE encodings, and even the max correlated component in the latent space is insignificant (Fig. 1d and Supplementary Fig. 1b). Controlling for one feature does not significantly impact the other dominant transformation features (i.e., polar orientation). With a double transformed output correcting two features simultaneously, we see decorrelation of both dominant features (Supplementary Fig. 1c), but the reconstructed images are poor (Supplementary Fig. 1g) reflecting the model's failure to learn relevant feature embeddings. The VAE with transformed output is shown to work on simple transforms such as rotation, but pairs of complex transformations like rotation combined with polar orientation prove too difficult. Both the β-VAE[21] and invariant C-VAE[22] also show strong correlations between the uninformative features we wanted to ignore and the latent space (Fig. 1e, f and Supplementary Fig. 1d, e). When both uninformative features are controlled for using the proposed ME-VAE with transformed image pairs, we see a decorrelation in both uninformative features, indicating that the VAE reconstructions learned to overcome them and focus on underlying features that better separate cell populations (Fig. 1g and Supplementary Fig. 1f). Unlike with the corrected output VAE, the ME-VAE produced coherent reconstructions (Supplementary Fig. 1i). Moreover, the ME-VAE is generalizable and scalable since it controls many uninformative features together in parallel by using a multi-encoder network where any number of encoders can be added, and each encoder learns a single transformation. Finally, when training on the same dataset of 15,898 single-channel images, all comparison architectures took a similar amount of computation time to train ranging between 53 and 54 s per epoch on average. The proposed architecture only took a few seconds longer, averaging 64 s per epoch, indicating that the increased performance and reduction in uninformative features do not come with a significant increase in computation time.

**Improving biological interpretation on single-channel images**. To evaluate the models' ability to improve downstream usefulness and biological relevance, we analyzed a dataset (see "Methods—

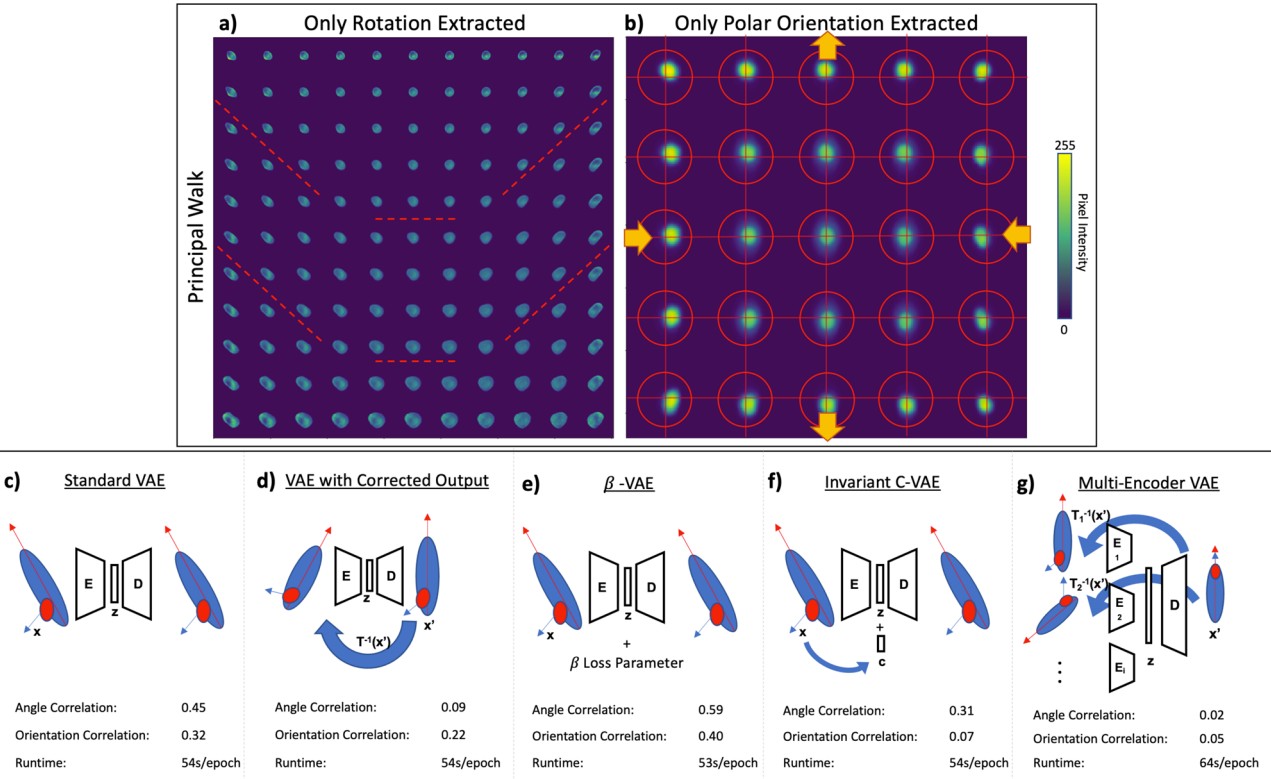

**Fig. 1 VAE hypersensitivity and proposed model architecture.** VAE analysis of two datasets is shown, each governed by a single biologically uninformative feature **a** rotation and **b** polar orientation. Principal walk reconstructions[23] show the VAEs' governing features across the latent space through a range of image reconstructions. To correct this model hypersensitivity, several architectures were tested: **c** standard VAE with matched raw images; **d** VAE with paired randomly transformed input and controlled output images; **e** β-VAE that operates similarly to the Standard VAE, but utilizes a β hyperparameter in the loss function to encourage an independent latent space; **f** an invariant conditional VAE that injects the values of the uninformative features into the decoder such that they are not embedded in the latent space; **g** the proposed multi-encoder VAE: VAE with corrections for multiple features (rotation, polar orientation, size, shape, etc.) using parallel encoder models, a shared latent space, and a single decoder model. In **c–e**, a correlation between the embedding components and the respective feature (angle and orientation) is measured to quantify how effectively the model removes uninformative features. PBS and TGFβ + EGF cell populations with single channels were used in this analysis ($n = 15{,}898$ single-cell images).

Datasets") of single-cell CYCIF images from MCF10A non-malignant breast epithelium cell line. The full dataset we analyzed is comprised of 6 ligand-treated cell populations and is stained with 23 biomarkers (Supplementary Table 1). Here we restricted our analysis to PBS (control) and TGFβ + EGF population and considered only the epidermal growth factor receptor (EGFR) channel. These were chosen because they have similar distributions of cell size and mean whole-cell EGFR intensity following cell-level normalization, making them difficult to naively separate with classical cellular features (Fig. 2a), but qualitatively show phenotypic differences such as compartment localization and stain texture. Within this dataset, we show that the ME-VAE better separates PBS- and TGFβ + EGF-treated cell populations compared to the standard VAE.

As can be observed in Fig. 2b, the standard VAE is incapable of separating the two cell populations, creating a mix of the labeled cell populations in k-means cluster space (number of clusters = 2) and UMAP embedding space. The cells within UMAP regions also have an arbitrary range of phenotypes; the only observed patterns are of uninformative features such as rotation, polar orientation, and size. Classically extracted features (Supplementary Table 2) show similar results to the standard VAE (Fig. 2c) where uninformative and non-biological features govern the clustering and UMAP distribution. Despite the fact that orientation was not included in the set of extracted properties, the rotation angle is still captured because the same information is available through a combination of important features such as eccentricity, extent,

moments, and inertia which were extracted. A denoising autoencoder, which is architecturally the same with no regularization loss on the latent space, performs similarly to the standard VAE with corrected output, having poor image reconstructions as a result of using transformed images that it has to overcome in a single encoder (Supplementary Fig. 1h). A result of this is that the denoising autoencoder reconstructs size as the primary feature with little other information capable of separating the cell populations (Fig. 2d). The β-VAE architecture[21] does not show significant improvement from the standard VAE either (Fig. 2e). Moreover, the β hyperparameter is known to be difficult to tune which can lead to large variations in both reconstruction quality and clusterability (Supplementary Fig. 2). The invariant C-VAE adapted from Moyer et al.[22] does see an improvement in clustering compared to the standard VAE (Fig. 2f), but despite having the uninformative values injected into the model, it is unable to keep them from being encoded in the latent space, resulting in UMAP embeddings dependent of uninformative features. Many of the recent extensions of the VAE that seek to improve the interpretability of the latent space simply modify the loss function used during training to encourage a result instead of forcing it (see "Methods—VAE models"). Unlike these previous attempts, the ME-VAE changes the actual deep learning architecture by adding multiple encoding blocks each for the purpose of removing a specific feature, making the implementation more straightforward for users (without introducing additional hyperparameter tuning such as β-VAE), and demonstrates increased performance.

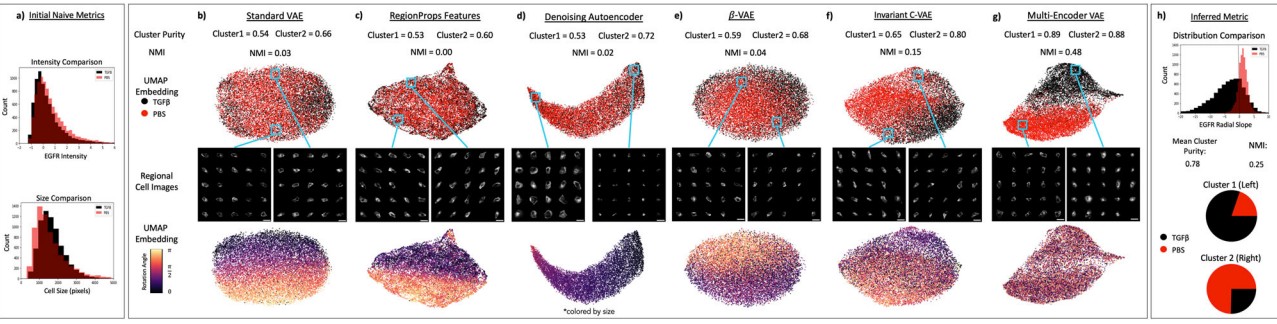

**Fig. 2 Separation of biologically distinct cell populations. a** Cells are compared using initial naive metrics such as mean EGFR intensity and cell size to show the difficulty separating the cell populations. **b–g** The model architectures are quantitatively evaluated using cluster purity and normalized mutual information (k-means with the number of clusters = 2). The sample size for all comparison methods and metrics is n = 15,898 single-cell images. A qualitative comparison is made using visual separation of two labeled cell populations in UMAP embedding space and visual analysis of cells from UMAP regions to identify biologically distinct factors. Rotation angles of cells are shown in UMAP embedding to show the influence of unimportant features on downstream analysis. **h** The same population of cells is compared using radial slope (a metric inferred from visually analyzing the regional cell images in **g**). Scale bars in **b–g** represent 20 μm.

By comparison to all other attempted methods, the ME-VAE has a dramatic increase in k-means cluster purity and Normalized Mutual Information and shows a clear separation of labeled cell populations in UMAP (Fig. 2g), indicating improved cluster-ability and separability. Regional cell images within the multi-encoder's UMAP space show distinct phenotypic differences that separate the cell populations with biologically relevant features (stain localization and subcellular pattern). In PBS dominant regions, EGFR stain is most heavily concentrated uniformly along the cellular membrane, while TGFβ + EGF regions show a cloudy diffuse concentration of EGFR stain throughout the cell with the heaviest concentration of stain localizing to one side of the nuclear membrane. These observed features illustrate a clear difference in cellular regulation and compartmentalization of the EGFR protein induced by the TGFβ + EGF ligand combination. Based on observations from the multi-encoder output in Fig. 2g, we were able to infer a metric of the radial slope which similarly separates the two populations (see "Methods—Evaluation metrics" and Supplementary Fig. 3). We observe a larger (more positive) radial slope in the PBS population on average, indicating that the distribution of stain increases radially toward the membrane, and by comparison the radial slope of the TGFβ + EGF population has a smaller (more negative) radial slope than the PBS, indicating that the stain distribution is located primarily toward the center of the cell and decreases radially toward the membrane. Using this metric, we see improved separation, cluster purity, and normalized mutual information compared to the selected naïve metrics (Fig. 2a, h). Furthermore, the concentration of EGFR in the TGFβ + EGF population is located just outside the nucleus, and therefore would not be successfully separated simply by isolating the mean intensity of the nuclear region. This demonstrates that ME-VAE yields biologically meaningful representations by identifying previously unappreciated features and generates biological discoveries by going beyond the limited set of known features such as mean intensity. The clustering metrics from the radial slope, however, are still lower than the full ME-VAE cluster purity, indicating more features beyond the radial slope are being extracted from ME-VAE.

**Use case with a large complex dataset.** Models were next trained on the expanded dataset (five ligands and PBS control) and 23 channel CYCIF images (see "Methods—Datasets" and Supplementary Table 1). Like before, the ME-VAE was trained to control for rotation, polar orientation, and cell size/shape. The standard VAE performed similarly to the previous experiment,

encoding cells based primarily on the dominant features such as size and rotation while largely ignoring complex staining information (Fig. 3). Although visually there is some preferential localization in UMAP (OSM left side, TGFβ + EGF right side), it is clear that the populations are thoroughly mixed with poor separability. The intensity profiles show that size has a strong impact on this left/right embedding (Fig. 3b). Most stains show little or no consistency within the embedding space, with the exception of DAPI and Ki67. These stains, however, show the same left/right distribution as size, indicating the nuclear intensity distributions are simply a result of cell size since the whole-cell mean intensity of a nuclear marker will decrease with larger cells and increase with smaller cells.

Despite the increased complexity of the multi-channel CYCIF images and a heterogeneous large dataset that could overload a simple architecture, the ME-VAE shows good separation of the labeled cell populations (Fig. 3a). We also observe subcluster formation for HGF, BMP2 + EGF, and TGFβ + EGF. By analyzing intensity profiles and regional cell images of these populations, we can see differences in expression (Fig. 3b and Supplementary Fig. 4b). The UMAP intensity profiles show clear stain intensity patterns indicating that the ME-VAE encoding space contains relevant biological information. Size does show some distribution in the UMAP, but the effect is largely dulled in comparison to the standard VAE.

Here we discuss some of the most noticeable drivers of separation between cell populations in the MCF10A dataset. PBS shows a marked decrease in Ki67 expression compared to other ligands, consistent with a relative decrease in proliferation. The TGFβ + EGF populations show an increase in S6 expression, indicating an increase in cell growth. This is observed visually with regional cell images (Supplementary Fig. 4b); however, it's worth noting that high S6 expression is seen in both large and small cells treated with TGFβ + EGF. In EGF- and BMP2 + EGF-treated populations, decreased expression of membrane adhesion proteins such E-cadherin and β-Catenin is observed. This decrease presents visually as a dim stain, but the marker is still localized to the membrane rather than missing or diffuse throughout the cell.

In both TGFβ + EGF- and PBS-treated cells, we see an increased concentration of HES1 localized primarily to the nucleus, while in other populations the distribution is uniform throughout the cell. In the case of TGFβ + EGF, this localization is accompanied by increased intensity (Supplementary Fig. 4b), but PBS intensity is more similar to the other ligand-treated populations. Similarly, Stat1a is primarily located in the nucleus

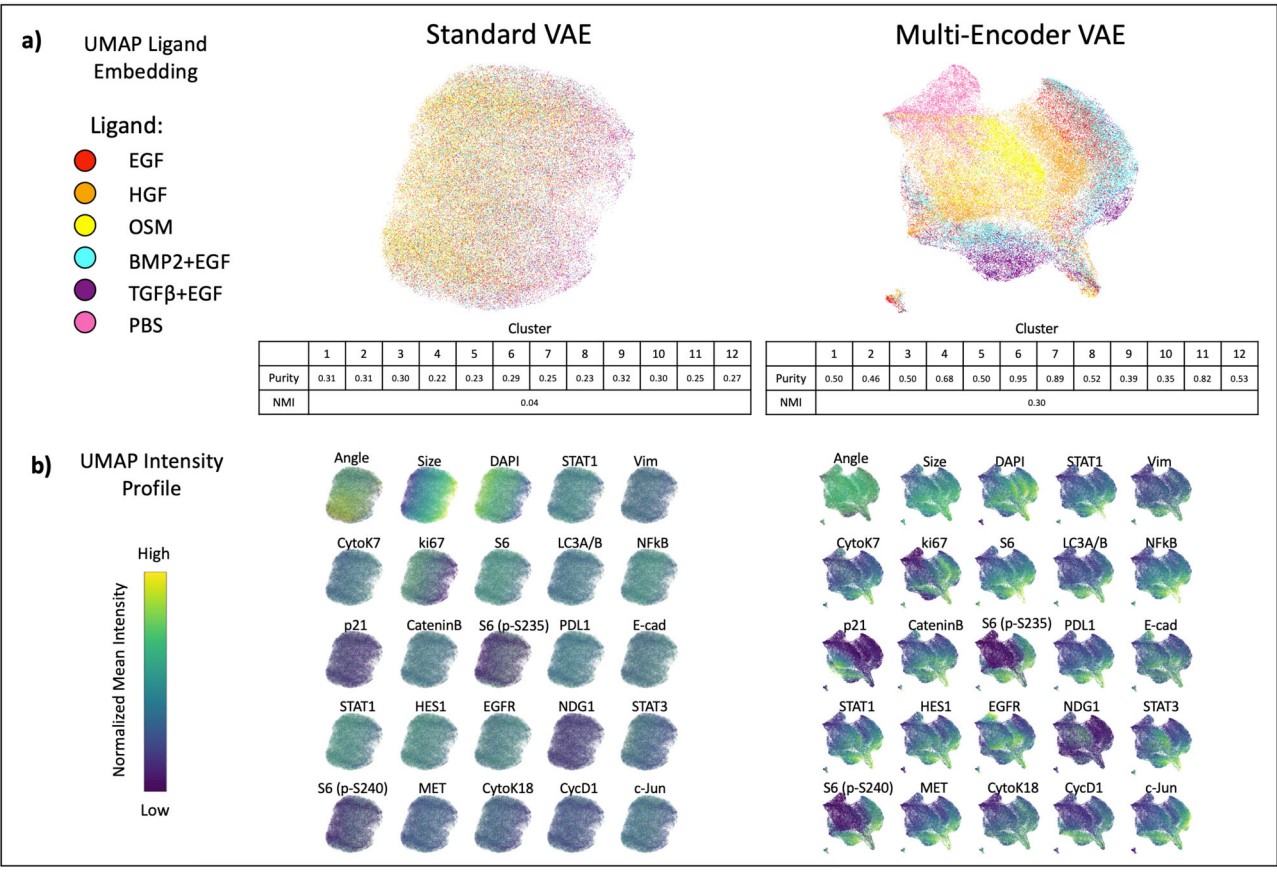

**Fig. 3 Ligand separation and feature distribution in full MCF10A dataset. a** UMAP embeddings for respective VAE encodings, allowing for qualitative visual evaluation of ligand separability. Cluster purities and normalized mutual information were calculated to quantitatively compare methods (*k*-means clusters = 12 to allow for ligand subpopulations). The mean cluster purity of the standard VAE was 0.04 with a standard deviation of 0.03. The mean cluster purity of the ME-VAE was 0.59 with a standard deviation of 0.15. The total sample size is $n = 73,134$ single-cell images. **b** Distribution of stain features across UMAP space, colored by intensity.

for TGFβ + EGF-, BMP2 + EGF-, and OSM-treated populations, but shows decentralized staining in cell images for other ligand populations. This is important because both HES1 and Stat1a are functional in the nucleus (Stat1 particularly as it translocates into the nucleus as part of its functional pathway) with limited activity in the cytosol[24,25]. Another observation is that p21 uniquely separates subpopulations in TGFβ + EGF-, HGF-, and BMP2 + EGF-treated cells, indicating that there are subsets of the population that are undergoing growth arrest due to inhibition of cell cycle progression via p21 regulation.

These results show that the ME-VAE captures relevant biological information and separates cell populations, highlighting important features without significant interference from the controlled uninformative features. Furthermore, the ME-VAE can capture emergent and biologically relevant imaging features not obvious without prior knowledge. By contrast, little to no biologically relevant information is obtained from the standard VAE.

**Correlation of reverse phase protein arrays pathway activity and CYCIF using ME-VAE features**. To validate that ME-VAE yields biologically more meaningful representations, we correlate VAE features with respect to Reverse Phase Protein Arrays (RPPA) pathway activity. By reordering VAE features using hierarchical clustering to form feature spectra, we extract broad patterns and reduce the dimensionality of the feature set. The standard VAE shows very poor self-correlation with only a

handful of feature clusters showing strong correlation (Supplementary Fig. 5a). Comparatively, we observe a clear pattern of self-correlation between ME-VAE features, indicating the model successfully extracts distinct yet different expression patterns (Fig. 4a). We identify ten representative clusters from the ME-VAE latent space that illustrate different expression patterns, which are explored using representative images (Fig. 4a). Representative cell images are chosen by selecting the cell for each feature set that has a high mean expression of all features in that respective aggregated feature set. Between clusters 0 and 1, we see a difference in the ratio of nuclear size and cell size. Cluster 1 encodes for larger nuclei than cluster 0 (this pattern is reaffirmed in Fig. 4b where cluster 1 correlates to DNA pathways and nuclear stains while cluster 0 does not). Cluster 4 is a highly varied cluster but contains large cells with more diffuse intensity patterns. From these aggregated features, we see that the ME-VAE architecture extracts a combination of intensity and morpho-spatial profiles with at least 10 clear axes of variation. Using these aggregated features, we can analyze and interpret biological meaning with fewer spurious correlations than comparing many to many.

A growing method for single-cell analysis is to integrate multiple modalities. Multi-modal integration helps validate where the two modalities overlap, expands the dataset with mutually exclusive or orthogonal features, and allows for cross-wise mapping of features. This is important for VAE based single-cell image analysis because it frames inherently obscure encoding features in a biological context and validates that the extracted

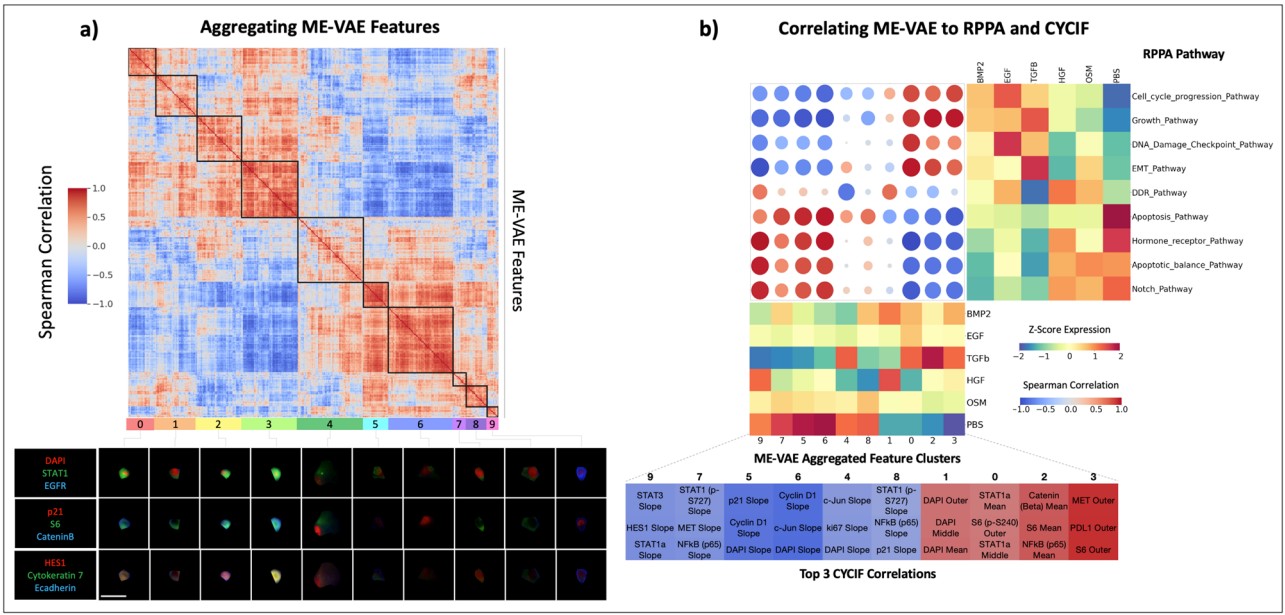

**Fig. 4 ME-VAE feature aggregation and transitive inter-modality correlation. a** Using the single-cell observations as features, correlations are drawn between pairs of ME-VAE features. These features are then hierarchically clustered to observe patterns and reduce VAE features to aggregated feature sets. Cell images were assigned aggregated feature scores using the mean expression of each feature in a cluster. Shown are representative cells that are highly expressing for each respective cluster. Scale bar represents 20 µm for all single-cell images. **b** Correlation matrix between RPPA pathway activity scores and ME-VAE aggregated features. Samples from the two modalities were paired by their ligand treatments, resulting in a sample size of $n = 6$ biologically independent ligand treated cell populations. RPPA pathways and VAE features were hierarchically clustered to show prominent patterns in correlation. ME-VAE aggregated features were also correlated to several metrics of CYCIF expression (mean inner, mean middle, whole-cell means, and radial slope) for all 23 stains. This CYCIF correlation was done using the full dataset of single-cell images (sample size $n = 73,134$ single-cell images). The table of CYCIF correlations shows the top three correlations for each ME-VAE aggregated feature.

features are coherent. The increased feature range of ME-VAE allows for cross-wise mapping and integration of complex CYCIF image features and other modalities (e.g., RPPA).

When correlating the seven aggregated standard VAE features with RPPA pathway activity, we notice two distinct issues. First, there is a single aggregated feature that shows significant correlations to nearly every RPPA pathway activity profile (Supplementary Fig. 5b). Second, there is a single RPPA pathway that correlates to nearly every standard VAE aggregated feature. When correlating standard VAE aggregated features to the extracted CYCIF metrics ("Methods—Evaluation metrics"), the Spearman correlations are small despite the increased sample size of $n = 73,134$ single-cell images (Supplementary Fig. 5b), with the largest correlations being restricted to nuclear markers such as CyclinD1, DAPI, and Ki67. As mentioned above this is likely an artifact of encoding for size since nuclear expressions can be a function of cell size. By contrast, the ME-VAE features result in more powerful and informative Spearman correlations with both RPPA pathways and CYCIF (Fig. 4b). All 10 aggregated features show strong and consistent Spearman correlations, illustrating that the ME-VAE has biological interpretability in both CYCIF and RPPA. Improved correlations demonstrate that the learned features from ME-VAE are biologically meaningful and illustrate the multi-encoder's applicability for multi-modal integration and comparison by extracting biologically meaningful features.

Biological correlations are validated by looking at representative images for each ligand treatment (Supplementary Figs. 4 and 6), where the stains shown were selected for their high correlations to the ME-VAE features or distinct visual patterns. The same patterns observed in the CYCIF correlation table and ME-VAE Z-score expression matrix (Fig. 4b), are also qualitatively confirmed by visual inspection. For example, S6 expression (ME-VAE aggregated feature 0) is high in BMP2 + EGF, EGF,

and TGFβ + EGF and is low in HGF, OSM, and PBS. Radial CyclinD1 radial slope (ME-VAE aggregated feature 6), as shown in Supplementary Fig. 6, is negative in BMP2 + EGF, EGF, and TGFβ + EGF, with high stain intensity in the inner compartment and rapid decrease toward the cell perimeter; conversely, HGF, OSM, and PBS show much dimmer CyclinD1 expression in the inner compartment. This pattern is even more clear in the radial HES1 slope (Supplementary Fig. 6, fourth column), where HGF, OSM, and PBS show a more continuous stain abundance all the way to the cell membrane. Although the RPPA sample size ($n = 6$ independent ligand treated cell populations) is still too small to achieve statistical significance, the correlations between protein markers in CYCIF and RPPA pathways linked by VAE features, are supported by known literature. DAPI expression (ME-VAE aggregated feature 1) is highly correlated to the DNA damage and repair pathway, which is expected since DAPI is a marker for DNA expression. A more interesting finding (ME-VAE aggregated feature 9) shows a strong correlation between the Stat3 radial slope of distribution and the epithelial-to-mesenchymal transition and hormone receptor pathways in RPPA. Prior literature also shows that Stat3 distribution throughout the cell, its translocation to the nucleus, and its cytoplasmic activation are important in the EGF-induced epithelial-to-mesenchymal transition pathway[26]. The ME-VAE architecture also extracts patterns when multiple markers play a role; CyclinD1 and p21 (ME-VAE aggregated feature 5) are known in the literature to play a joint part in the cell growth pathway[27]. These observations demonstrate a potential application of multi-modal integration using the proposed approach for other single-cell image analysis[28].

The ME-VAE can also improve downstream analysis by increasing population separability (Fig. 5) as measured by mean pairwise Tukey $p$ values and mean effect sizes. For the given MCF10A dataset, the CYCIF marker panels were chosen with the

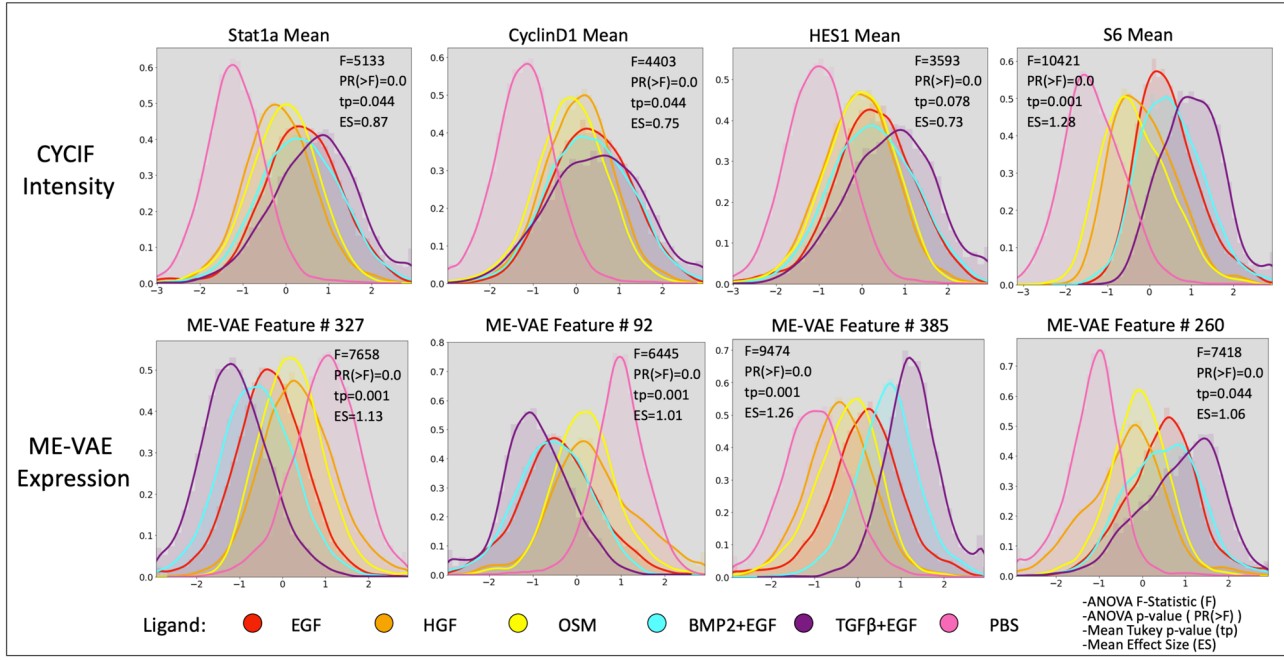

**Fig. 5 Separability of ligands using individual ME-VAE features.** Density function for several CYCIF and ME-VAE feature pairs. A two-sided ANOVA was performed for features and intensities between populations in order to compute the $F$ statistic and $p$ value (PR(>$F$)). Subsequently, the mean Tukey pairwise $p$ value across ligands and mean effect size are shown for each feature. ME-VAE features used for comparison were the features with the largest correlation to the respective CYCIF marker. This analysis utilized all 73,134 cell images from the MCF10A dataset.

known ligands and cell populations in mind to highlight differences between the populations and separate them. This results in already decent separability using just CYCIF mean intensity information (Fig. 5). That being said, ME-VAE features show lower mean Tukey pairwise p-values indicating a greater average significance in separability, and the effects sizes for those separations are larger (Fig. 5). The marker that was an exception to this (shown in the first example) is S6, where the CYCIF mean intensity shows better separability. Even in this example, however, the multi-encoder's feature is still adequate. It is worth noting that ME-VAE latent space features are encoded in combination to represent even a single stain, so separability can be improved even further when utilizing more than just one feature at a time.

Although aggregated features are useful for integrating data modalities, since they reduce spurious correlations, using the full range of latent features is preferable for clustering populations since aggregation can average out some relevant signal (Supplementary Fig. 7). The aggregated features still perform well at separating cell populations and still outperform most stains, however, there is a noticeable reduction in effect size after aggregation.

## Discussion

Just as it is necessary to pre-process, normalize, and remove unwanted features from scRNAseq or RPPA analysis, so too is it necessary to remove uninformative features from single-cell imaging analysis in order to extract features of interest. Without this guided feature alignment, VAE applications for single-cell image analysis will only reconstruct dominant features while ignoring subtle more informative features (Fig. 1a, b). By making uninformative features invariable across a dataset using pairs of transformed images in parallel encoding blocks (Fig. 1g), VAE priority can be shifted to mutually shared, biologically relevant information (Figs. 2g and 3). This results in a more complex and meaningful latent space.

Feature extraction is important for all downstream analysis and interpretation, but oftentimes naïve metrics are not sufficient to capture biological differences and separate cell populations, especially in datasets where labeled populations are not known beforehand. By separating populations with the ME-VAE, distinct populations and biologically meaningful metrics can be established, allowing identification of emergent image properties such as localization and staining pattern (Figs. 2g and 3 and Supplementary Fig. 4), with increased separability compared to using intensity or morphology information alone (Figs. 2c, g, h and 5). Although a theoretically infinite number of handcrafted naïve features could be designed to capture more information, the advantage of deep-learning is that it can extract the most important features of an image with limited prior knowledge required. More complex single-cell analysis methods such as multimodal integration (Fig. 4) require a wide range of biologically relevant features. The ME-VAE architecture provides an important step for biological research by linking imaging data to molecular readouts. By employing this architecture to extract a larger range of features and metrics from single-cell images, potential applications, such as multi-modal integration using imaging features, become available that were previously restricted due to inadequate cell representations.

Generalizability of the model was evaluated (Supplementary Fig. 8) by testing on an unseen replicate of MCF10A cells treated with two ligand perturbations. The ME-VAE is able to separate the cell populations with similar efficacy without having seen the cell images during training. To further demonstrate the generalizability of the ME-VAE architecture with a large complex dataset, we applied ME-VAE to another multiplexed imaging modality, CODEX (see "Methods—Datasets" and Supplementary Table 3). The same overall increased performance is observed in the additional dataset of single-cell images extracted from CODEX tissue microarrays (TMA)[29] (Supplementary Fig. 9), where the Standard VAE mixes populations and organizes cells primarily based on size. By contrast, the ME-VAE forms distinct

clusters with unique expression profiles and is even able to extract cell types with known size differences, for instance, macrophages (as determined by high CD68 expression). In the CODEX dataset, the ME-VAE was only correcting for rotation and polar orientation, since size and shape were considered to be more biologically relevant variables of interest in this setting. This illustrates the ME-VAE's ability to generalize to additional modalities, cell types, as well as to tissue data.

The simplicity of the multi-encoder design makes it easily incorporated into other deep-learning architectures that have other advantages, such as being augmented with a discriminator to improve reconstruction quality. An example of this is shown by integrating a denoising autoencoder with the multi-encoder architecture with self-supervised signal which encourages better reconstructions at the cost of a less interpretable latent space (Supplementary Fig. 10). The clustering results of the multi-encoder denoising autoencoder (ME-DAE) are slightly worse than the ME-VAE, suggesting that the denoising aspect improves reconstruction but makes the latent space less useful. The ME-VAE methodology is limited by two criteria which the uninformative features must meet: (1) being known so that they can be addressed with a new encoding block and transformed image pair; (2) being a known or inducible transform operation such as rotation, affine, or scale such that a respective randomly transformed image can be generated using the operation. Despite these limitations, the majority of dominant uninformative imaging features are based on known transformations, making the ME-VAE architecture widely applicable.

Computationally the model is not significantly larger than a standard VAE or other comparable architectures required for training (Fig. 1c–g), excluding the time necessary to transform images which will vary based on transformation complexity, as it only adds a single encoding block for each undesired feature. Future applications of this architecture will allow complex features such as texture, pattern, and distribution to be extracted from single-cell images without the hassle of disentangling dominant uninteresting transform features. Images contain morpho-spatial features not shared by their other single-cell counterparts (scRNAseq and RPPA), and by implementing this architecture, the scientific community will be able to analyze these unique image features with the same robustness as algorithms made for other well-established single-cell modalities.

## Methods

**Datasets**. MCF10A cell populations were treated with seven ligands, PBS (control), HGF, OSM, EGF, BMP2 + EGF, TGFβ + EGF, and EGF + IFNγ (data from the LINCS Consortium—https://lincs.hms.harvard.edu/mcf10a/)[30]. For this paper, we analyzed all but the IFNγ population because the initial analysis showed that it was so distinct from other cell populations that even a single marker intensity resulted in decent separability. After 48 h, cells were fixed and subjected to cyclic immunofluorescence with 23 markers shown in Supplementary Table 1. The dataset comprises three plates of replicates. On each plate, there are three replicates of each ligand in different wells, and in each well nine different fields of view were taken. Cells were segmented using the Cellpose segmentation tool[31] using the EGFR and DAPI channels. Stains were normalized using histogram stretching to the 1st and 99th percentiles across intensities for individual cells and across the whole dataset. Image transformations were applied for rotation, polar orientation, and size/shape (Supplementary Fig. 11). Rotation was corrected by obtaining the major axis from the binary cell mask, then rotating the image using the Python package OpenCV[32]. Polar orientation was corrected by calculating the angle toward the image's center of mass, then applying a flip/rotation to align the angle using the Python NumPy package[33]. Size/shape was corrected simultaneously by registering the cell mask to a circle target image (code available here: https://github.com/GelatinFrogs/Cells2Circles). In total, 73,134 cells were processed through this pipeline. When isolating the PBS and TGFβ + EGF populations for the two-ligand separation analysis ("Results—Controlling for uninformative features and Improving biological interpretation on single-channel images"), the sample size was 15,898. All 73,134 cells were analyzed in the full MCF10A analysis, modality integration, and separability test ("Results—Use case with a large complex

dataset and Correlation of reverse phase protein arrays pathway activity and CYCIF using ME-VAE features").

A publicly available CODEX dataset (the patients' consent to use their tissues for research)[29] was used as a secondary multiplex imaging technology to demonstrate the generalizability of the ME-VAE to other tools, cell types, and to tissue data. The portion of the dataset tested consisted of eight TMAs from the skin and breast cancer. From the full panel of 91 markers, we chose 20 stains that were the least sparse, highest quality, and important for labeling the full tissue (Supplementary Table 3). We then segmented 12,229 cells from the TMA images using the Hoechst and CD71 channels in Mesmer[34], and normalized using histogram stretching to the 1st and 99th percentiles across the whole dataset.

Bulk Reverse Phase Protein Array (RPPA) was performed by the LINCS consortium[30] in parallel to the CYCIF imaging, on cell populations treated with the same ligands after 48 h of exposure. The protein array incorporated 295 protein markers. RPPA data were median-centered and normalized by the standard deviation across all samples for each component to obtain the relative protein level[35]. The pathway score is then the sum of the relative protein level of all positive regulatory components minus that of negative regulatory components in a particular pathway. Pathway members and weights were developed through a literature review. Pathways were used instead of individual proteins because a large number of proteins would decrease the significance of correlations. Despite the available bulk RPPA dataset having a smaller sample size than the single-cell CYCIF dataset, it was chosen as the secondary modality because similar ligand separation and cluster patterns were observed in both modalities, indicating an overlap in the information each contains (Supplementary Fig. 12).

For correlation to CYCIF and RPPA pathways, the VAE latent space was restricted to smaller sets of aggregated features. These aggregated features were made using self-correlation of VAE features across individual cell metrics and averaging the VAE features for resulting hierarchical clusters (Fig. 4 and Supplementary Fig. 5). This was done to reduce the feature dimensionality and reduce spurious correlations in the biological findings. Representative images for each cluster were done by finding cells with a high average expression for all features within the cluster. For other analyses of VAE features comparing VAE separability to CYCIF expression and interpreting image feature space, ME-VAE encoding features were restricted to 18 single features for each. The dimension of 18 was chosen because it is the number of mutual markers between the RPPA and CYCIF datasets. Explanatory features were chosen from the VAE encodings such that the inter-cluster variability was maximized and the intra-cluster variability was minimized using the following equation:

$$\text{Feature score} = \text{Var}_{\text{all}} - \frac{\sum\limits_{i=\text{cluster}} \text{Var}_{C_i}}{\text{\# of clusters}} \qquad (1)$$

**VAE models**. To allow for a fair comparison, the structure of the encoder and decoder blocks were kept consistent between networks, and the same latent dimension was used for all models for a given dataset (64 for the 1-channel dataset, 512 for the 23-channel dataset). Both encoder and decoder blocks consist of three layers, and all layers utilize a Rectified Linear Unit activation except the final output, which uses sigmoid activation. All models were trained for 10 epochs (determined by identifying the loss function plateau) on the NVIDIA P100 with 100 GB of RAM and 100 GB of disc space, but the ME-VAE architecture can work on any NVIDIA GPU.

A standard VAE with matching pairs of single-cell images was used to establish baseline performance (Fig. 1c). Standard VAEs utilized the standard Evidence Lower Bound loss format characterized by reconstruction and Kullback–Leibler (KL) divergence terms. We used a Binary Cross-Entropy loss (BCE) as the reconstruction term for all VAEs tested here to keep the comparisons fair and consistent. Put together, the standard VAE loss used was:

$$L_{\text{Standard VAE}} = \text{BCE}(x, p(z)) - \text{KL}(q(z|x)||p(z)) \qquad (2)$$

where $q$ represents the encoder and $p$ represents the decoder as described in Kingma et al.'s initial VAE paper[9]. Here $x$ represents the unadjusted input image and $z$ represents the latent space.

By using an image randomly transformed with respect to a dominant feature as the input and controlling for the same uninformative feature in the output image (Fig. 1d), the model can self-supervise the transformation and will only encode novel features since the controlled features (such as rotation) no longer aid reconstruction:

$$L_{\text{Output Corrected VAE}} = \text{BCE}(x', p(q(z|T^{-1}(x')))) - \text{KL}(q(z|T^{-1}(x'))||p(z)) \qquad (3)$$

where $x'$ represents an image that has been transformed with a known transformation to remove one or more uninformative features and $T^{-1}(\cdot)$ represents a transformation of the controlled image to create a dominant uninformative feature at a random degree.

The proposed multi-encoder architecture uses multiple transformed inputs with separate encoder blocks, where each block controls for a separate uninformative feature, and a single decoder block uses the shared latent space (combined by multiplication to emphasize mutual information) for reconstruction (Fig. 1g). To accommodate the multiple encoders in the loss, the KL term is replaced with a summation of all KL divergences for each individual latent space, which is then

divided by the total number of encoders ($n$):

$$L_{\text{ME-VAE}} = \text{BCE}(x', p(z_{\text{all}})) - \frac{1}{n}\left(\sum_{i=1}^{n} \text{KL}(q_i(z_i|T_i^{-1}(x'))||p(z_i))\right) \quad (4)$$

where each encoder's ($q_i$) individual latent space ($z_i$) is combined in an element-wise multiplication layer to create a mutual latent space ($z_{\text{all}}$) and $T_i^{-1}(\cdot)$ represents a different random transformation for individual uninformative features such as rotation, polar orientation, size, shape, etc., respectively. The shared latent space of the multi-encoder forces the deep learning model to encode features that are shared between each transformation, reinforcing the shared mutual information and eliminating the non-shared transformational information. A base implementation of the ME-VAE architecture can be found here: https://github.com/GelatinFrogs/ME-VAE_Architecture. The multi-encoder architecture allows for image pairs to be randomly transformed, which can act as a balancing agent for imbalanced features. Furthermore, the corrected outputs serve as a weakly self-supervised signal for the model. With the extra information from the additional inputs, the model is able to overcome more complex transformations that failed in the corrected output architecture in Fig. 1d, when multiple corrections are attempted. Paired images also serve one additional benefit of allowing for features to be retained in parallel encoders that might be lost due to artifacts in other corrections, i.e., artifacts within a polarity correction encoder will not be present in a rotation correction encoder.

The β-VAE[21] makes a small but significant change to the standard ELBO loss function of the Standard VAE by adding an adjustable hyperparameter to the loss function:

$$L_{\beta\text{-VAE}} = \text{BCE}(x, p(z)) - \beta * \text{KL}(q(z|x)||q(z)) \quad (5)$$

where the β applies varying amounts of priority to the KL regularization term. This forces the VAE to separate features into a more interpretable format where each component corresponds to a specific feature[21]. One downside of this method is that shifting priority to the regularization term causes the model to produce poorer quality reconstructions since less priority is placed on the reconstruction term. Another downside is that the β hyperparameter can be difficult to tune properly, and a different β value will be optimal for different datasets, image sizes, and latent dimensions (Supplementary Fig. 2).

A final architecture tested was the invariant conditional autoencoder (C-VAE), which injects a vector of quantified class/values of interest into the decoder (here termed $c$). The improvement this architecture makes upon the ELBO loss used by standard VAEs and conditional VAEs is the addition of a conditional and marginal KL regularization term that operates similar to a Maximum Mean Discrepancy penalty by pushing the latent spaces to be the same for varying $c$ values[22]. In our application this means that the resulting latent space will ideally be independent of the undesired values injected into the architecture through the c vector:

$$L_{\text{C-VAE}} = -\text{KL}(q(z|x)||p(z)) - \lambda\text{KL}(q(z|x)||q(z)) + (1+\lambda)\text{BCE}(x, p(z)) \quad (6)$$

where $\lambda$ is a hyperparameter which during this experiment was set to 1. The invariant conditional VAE was adapted based on the paper by Moyer et al.[22] and code available from the author's GitHub and tutorials (https://github.com/dcmoyer/invariance-tutorial/blob/master/tutorial.ipynb). In our implementation, we used the quantified values of rotation angle, polar orientation, and size as the c inputs such that the latent space would hopefully be invariant to those features. Similar to our proposed approach, the uninformative features of interest must be known and quantifiable in this method since the c values are input into the model.

Denoising autoencoder methods (Fig. 2d and Supplementary Fig. 10) used the same architecture as the standard VAE and the ME-VAE, respectively, but in the case of the standard denoising autoencoder the model was also trained using the transformed image sets. Denoising autoencoders do not use a regularization term in the latent space, relying solely on the reconstruction term for loss. As such, both methods used the loss function:

$$L_{\text{Denoising AE}} = \text{BCE}(x', p(q(z|T^{-1}(x'))))$$

Additionally, in the reduced dataset, the architectures are compared to classically extracted intensity and morphology features using scikit-image's regionprops package[36]. The classical feature dataset is defined by 58 properties (Supplementary Table 2) extracted from each single channel cell image. We included all properties we could for single-channel images but left out orientation in order to show that rotation angle is still captured through other properties even when it is not explicitly an extracted feature.

**Evaluation metrics.** In order to evaluate the model's ability to separate labeled cell populations, k-means clustering was applied to the encoding spaces using sklearn[37]. Cluster purity was then calculated by taking the percentage of the largest population for each cluster. UMAP embeddings were calculated using the UMAP Python package[38]. Regional cell images within UMAP (Fig. 2 and Supplementary Fig. 4) were created by sampling cells from various regions of the UMAP embedding space to give visual context to the features that are being separated. Cells were placed into regions by snapping their embeddings to a grid and taking one representative image from each point on the grid[23]. Biological metrics were calculated to give VAE encoding features biological grounding (Supplementary Fig. 3). Circularized cells were used for calculation because they made the

compartmentalization of the cell more consistent and uniform. Mean intensities were calculated for inner, middle, outer, and whole-cell compartments. To calculate the radial slope, the mean intensity was taken from each radius of the circularized cell, then the linear regression of the series was calculated using the scipy.stats package[39] in Python. The slope of the calculated linear regression was used as the metric and the intercept was ignored. Self-correlations between VAE features were performed using Spearman correlation and clustering was done in seaborn clustermap[40]. Clustermaps using hierarchical clustering were calculated using the method's default distance metric (Euclidean). Representative cluster images were chosen based on the high expression of the cluster's respective VAE features. RPPA pathways activity scores, VAE features, and biological metrics were all standardized prior to analyses using the sklearn StandardScaler function[37] in Python.

**Statistics and reproducibility.** To test whether a specific feature was embedding in the latent space, the max spearman correlation was taken between the feature and the components of the latent space ($n = 15,898$ cell embeddings). Spearman correlations between RPPA pathway activities and VAE encodings and between CYCIF and VAE encodings were both calculated using the Spearman correlation ($n = 6$ perturbation conditions). To test for separability (Fig. 5 and Supplementary Fig. 7), features were first tested using type 2 ANOVA with the Python implementation of anova_lm from statsmodels[41] for the default $F$-statistic, all of which proved significant ($n = 73,134$ single-cell images). Subsequently, the post hoc pairwise Tukey $p$ test was used to calculate the significance and effect size for each ligand pair ($n = 73,134$ single-cell images). The mean $p$ value and effect size are reported to illustrate average separability.

**Reporting summary.** Further information on research design is available in the Nature Research Reporting Summary linked to this article.

## Data availability

CYCIF and RPPA data are publicly available through the LINCS Consortium: https://lincs.hms.harvard.edu/mcf10a/. CODEX data are available online (https://doi.org/10.7937/tcia.2020.fqn0-0326). Supplementary Data 1: EGFR intensity, cell size features for Fig. 2a and EGFR Radial slope features for Fig. 2h.

## Code availability

For reproducibility, we share the code with precise implementation; further details describing variables and equations, as well as shared trained models with parameters in GitHub. All ME-VAE codes are available on GitHub (https://github.com/GelatinFrogs/ME-VAE_Architecture).

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

## Acknowledgements

This work was supported in part by the National Cancer Institute—U54HG008100 (Gray), U54CA209988 (J.G.), U2CCA233280 (J.G.), U01 CA224012, R01 CA253860 (Y.H.C.)—U01 CA217842 and 1U01 CA253472-01A1 (G.M.; Korkut; Liang), and the OHSU Center for Spatial Systems Biomedicine. The resources of the Exacloud high-performance computing environment developed jointly by OHSU and Intel and the technical support of the OHSU Advanced Computing Center are gratefully acknowledged.

## Author contributions

Conceptualization, L.T. and Y.H.C.; methodology, L.T. and Y.H.C.; software, L.T.; formal analysis, L.T.; RPPA analysis, M.D., M.L., and S.G.; writing—original draft, L.T.; writing—review and editing, L.T. and Y.H.C.; supervision, G.M., J.G., L.H., and Y.H.C.; funding acquisition, J.G., Y.H.C., and G.M.

## Competing interests

L.T., M.D., S.G., M.L., L.H., and Y.H.C. have no competing interests. G.M. is a SAB/Consultant for the following: Abbvie, Amphista, AstraZeneca, Chrysallis Biotechnology, GSK, Ellipses Pharma, ImmunoMET, Ionis, Lilly, Medacorp, PDX Pharmaceuticals, Signalchem Lifesciences, Symphogen, Tarveda, Turbine, Zentalis Pharmaceuticals; has licensed technology in the following: HRD assay to Myriad Genetics, DSP patents with Nanostring; and G.M. has stock, financial, or other interests in the following: Catena Pharmaceuticals, ImmunoMet, SignalChem, Tarveda, Turbine. J.G. has licensed technologies to Abbott Diagnostics; has ownership positions in Convergent Genomics, Health Technology Innovations, Zorro Bio, and PDX Pharmaceuticals; serves as a paid consultant to New Leaf Ventures; has received research support from Thermo Fisher Scientific (formerly FEI), Zeiss, Miltenyi Biotech, Quantitative Imaging, Health Technology Innovations, and Micron Technologies.
