## [Transparent Peer Review File · Communications Biology]

A Multi-Encoder Variational AutoEncoder controls Multiple Transformational Features in Single Cell Image AnalysisReviewer comments, first version:

Reviewer #1 (Remarks to the Author: Overall significance):

The paper titled "ME-VAE: Multi-Encoder Variational AutoEncoder for Controlling Multiple Transformational Features in Single Cell Image Analysis" by Luke Ternes and colleagues describes a novel computational model called Multi-encoder VAE (ME-VAE) for single cell image feature extraction that removes specified uninformative features by making them uniform and invariant across the reconstructions, using modified pairs of transformed input and output images by self-supervised transformation, and utilizing multiple encoding blocks. Using the ME-VAE to control for these multiple transformational features, the authors are able to extract biologically meaningful and transform-invariant single cell information and better separate heterogeneous cell types. The approach is novel, aims to address an important problem, and results in improved downstream results compared to the Standard VAE using no informed transformations. The authors also illustrate the ability of ME-VAE for multi-modal integration and comparison.

Reviewer #1 (Remarks to the Author: Impact):

I do think this is an important paper but it needs major revisions (as I detail below) and seems more appropriate for Nature Comp Sci or Comms Biology. However, if the authors make the changes suggested and do a great job it could be appropriate for Nature Comms.

Reviewer #1 (Remarks to the Author: Strength of the claims):

There are key limitations to this work, first the lack of details pertaining to generalizability and scalability, and the reduced clarity in presentation of the data, along with incomplete explanation of figures and equations. The manuscript feels rushed and not quite ready for submission, adding to the lack of clarity and readability.

I highlight other concerns below that I think should be addressed.

1. The first limitation is the lack of generalizability to other emerging multiplexed technologies such as CODEX, or MIBI.

As mentioned in the introduction, there are upcoming multiplexed imaging technologies. In the current work, the authors only show ME-VAE on CYCIF data. For generalizability of such novel methods, it is essential to demonstrate ME-VAE on one other imaging technology. There is public data available for both CODEX and MIBI. For example see:

<https://portal.hubmapconsortium.org/docs/assays/codex> and <https://www.angelolab.com/mibi-data>

2. In the last section of Results (A), the authors mention about generalizability and scalability. To address generalizability, please refer to comment #1. To address scalability, please show runtime benchmarks of ME-VAE against Standard VAE for one of the experiments (e.g. between Figure 1c - e)

3. Regarding known controllable transformations: The results are shown for features that are known controllable transformations. These are then used as self-supervision to extract invariant features during model training. What about the case of noise-induced transformations that are unknown? Further, some of the known uninformative transformations such as rotation and polar orientation are not independent features. How do we know that these uninformative features are not getting

mixed across encoders?

4. Size and shape of a cell are important and informative features. For example, depending on the tissue being imaged and the context, certain cell types are larger than others (e.g. macrophages), or they might have a certain shape (spindle-like). This information is essential to be able to segregate them. Is it then justifiable to convert these features to being uniform and invariant across transformations?

5. The crux of this work relies on transformed image pairs. What are these image pairs – an input image and its transformed output? Or are these the two transformed images, one for rotation and one for polar orientation?

6. Figure 2:

a. Legend says 'Rotation angle of cells are shown in UMAP embedding to show the influence of unimportant features on downstream analysis'. Where is this shown in the figure?

b. For Figure 2b, what is the input to k-means? Also mention what each dot is in the UMAP or k-means plot. How many dots are shown in the figure?

c. What are regional cell images (e.g. in Figure 2b, c)? The blue square seems to have many dots whereas the zoomed in regional cell image shows 25 cells. Please also provide one higher resolution color image, with an explanation of biologically relevant features (stain localization, intensity, and subcellular pattern) within this zoomed-in regional cell image.

d. What are the radial slopes for Figure 2c. Since this is computed by fitting a regression line, how can a same/similar slope distinguish similar distributions for different cell types?

e. 'The cluster purities from radial slope metrics, however, are still lower than the full ME-VAE cluster purity, indicating more features beyond the radial slope are being extracted from ME-VAE': Is this really a case of more features or is this a case of ME-VAE being overfit to the 'noise' that got extracted?

7. Figure 3: 'Size does show some distribution in the UMAP': Please highlight this in Figure 3, Supplementary 3b

8. In Figure 4a (bottom), each column is a cluster and is identified by a set of differentially expressed markers. Why is then each row showing a different set of differentially expressed markers per column? Same comment for Supplementary Figure 4

9. Please give an example of 'morpho-spatial profiles' (mentioned in Results D)

10. Supplemental Figure 4b: Please highlight or mention in the legend the row/column number where the following is observed: a 'single aggregated feature that shows significant correlations shows correlates to every RPPA pathway activity profile (Supplemental Figure 4b). Second, there is a single RPPA pathway that correlates to every standard VAE aggregated feature.'

11. In Results D, please add citations for 'known biology', 'known literature'.

12. In the Discussion, there is mention of 'augmenting' the model. What would an example for an augmented feature be and how would this be transformed for the ME-VAE

13. How reliable was the EGFR channel for segmentation? For cells where the EGFR signal is not clear, would it not help to identify such cells by using additional nuclear markers for segmentation? For the extended dataset, was the segmentation again done using only the EGFR channel? If only EGFR was used, why was this the case?

14. Figure 5: 'ME-VAE features used for comparison were the features with largest correlation to the respective CYCIF marker'. Why not compare CYCIF with the ME_VAE clustered (aggregate) features? The authors already point out that they do hierarchical clustering on the ME-VAE feature 'to reduce the feature dimensionality and reduce spurious correlations in the biological findings.

- a. This comparison would also give an idea of how the clustered features look like
- b. Further, how many ME-VAE and Standard VAE features were there?
- c. Is there any close correspondence between the z-scores in either column per row?

15. 'ME-VAE encoding features were restricted to 18 single features for each'. Does this mean that 1 ME-VAE feature = 18 single features? If this is the case, how were 18 single features assigned to one ME-VAE feature?

16. Equations in Methods B: Please explain all the variables and what the equations do.

17. Figure 1: Mention the data used, number of cells etc. in the Figure legend.

18. What are the data dimensions for the RPPA dataset?

19. There are two cell numbers reported – 71314 and 73,134. Is the former after pre-processing the images?

20. C. Evaluation metrics: Explicitly state how the slope was calculated: was it using the β from the regression equation?

21. Which clustering method was used from the seaborn clustermap function?

22. Please spell check the document. There are typographical errors relating to words e.g. decrease, separability, reconstruction, hierarchical, python, spearman, as well as word repeats.

23. Supplemental Fig 4: correct the text to reflect Standard VAE.

24. Figure 5: specify which type of ANOVA was used, and what was the p-value or F-statistic and depict this in a figure.

Reviewer #1 (Remarks to the Author: Reproducibility):

The authors do host the code on GitHub and provide appropriate documentation.

Reviewer #2 (Remarks to the Author: Overall significance):

Ternes et al. propose an extension of the classical VAE (variational autoencoder) for single cell image analysis for the purpose to extract biologically more meaningful latent representation of the input images. The main motivation is that the vanilla VAE tends to identify non-biological images features present in the dataset, such as rotation, scale etc, which can be viewed as confounding factors/ biases in the training dataset. The authors propose a method, called ME-VAE, to remove these non-informative features from the latent representation, hoping that the resulting new latent representation can lead to a better clustering or characterization of cell types/states.

The main idea behind ME-VAE is data normalization plus data augmentation. It generates a new set of target images that have been properly normalized, corrected based on a given set of predefined

transformations. It then trains the model with random transformations of the input images, forcing the model to learn to ignore these transformations and focus on biological more meaningful features. The authors demonstrated that ME-VAE was able to yield biologically more meaningful representations than VAE through clustering and correlation analysis.

Reviewer #2 (Remarks to the Author: Impact):

It seems to me a more focused, specialized journal is more appropriate for this manuscript.

Reviewer #2 (Remarks to the Author: Strength of the claims):

Although the work is potentially interesting, I have the following major concerns:

1. The authors focus on comparing ME-VAE to vanilla VAE. However, this is highly biased for several reasons. First, there are several other recent works on single cell image analysis that have not been properly discussed, and certainly not experimentally compared. I highly recommend the authors take a close look at the methods described in the following paper and carry out a thorough comparison analysis against these existing methods.

MCMICRO: A scalable, modular image-processing pipeline for multiplexed tissue imaging by Schapiro et al.

Second, going back to the VAE method itself, it is well known that VAE does not handle confounding factors well. There are many existing works on how to correct confounding factors on VAE. Some of these methods have also been proposed for single cell genomic data analysis. A few references include:

Moyer, D. et al. (2018) Invariant representations without adversarial training. *Advances in Neural Information Processing Systems*, 31, 9084–9093.

Deep Generative Modeling for Single-cell Transcriptomics, Romain Lopez et al, *Nature Methods*, 2018

Cao et al, SAILER: Scalable and Accurate Invariant Representation Learning for Single-Cell ATAC-Seq Processing and Integration, 2021

Although they are applied to different types of datasets, the methods themselves can be applied to single cell image analysis as well. Instead of comparing with vanilla VAE, the author should compare with these more recent extensions of VAEs.

2. It's also unclear to me why VAE is a good method for single cell image analysis. VAE is a generative model. The Gaussian prior applied on the latent variable tends to pull all representations toward the origin, and consequently reduces the separation between different cell types. The authors should provide a justification on why VAE is a good model for single cell analysis, and why it is better than a simpler denoise auto-encoder, the non-generative model.

3. The approach works for pre-defined, well-known confounding factors such as rotation, scale. But what about latent features not associated with a well-defined transformation? It is well known that deep learning models tend to pick up correlated features that are not biologically meaningful. How do you plan to handle these features, which are a) not known beforehand, and b) may not be

associated with a rigid simple transformation.

4. Because the current model doesn't address batch effect, the better clustering shown in Figure 2 can potentially be associated with the batch effect. I would recommend testing the model on biological replicates of the same cell types to show that cells of the same type from different batches are mixed.

5. I would also like to see the results from the samples not in the training dataset. If the features are truly biologically meaningful, I would expect to see similar results on these samples as well.

6. Regarding the method itself, the authors should compare with the vanilla VAE using normalized/corrected images, that is, applying VAEs on normalized images instead of raw images.

7. Please use standard metrics such as ARI, NMI to evaluate clustering qualities.

8. I also highly recommend the authors to test the method on a separate, ideally public dataset.

Some minor comments:

1. The description of VAE models in Method B should be substantially improved. Notations are non standard. Variables are often not defined or not referenced. Equations are unlabeled, and which loss function is for which model is not mentioned. Equation of L_e seems to use terms T_i^{-1} .

2. The ELBO of VAE contains a reconstruction term and a KL-divergence term encouraging smoothness of the latent space. The KL term seems to be missing from the loss functions.

3. Since the vanilla VAE uses isotropic Multivariate Gaussian for prior, the KL term will facilitate different dimensions of latent z to be independent with each other. Later proposed disentanglement methods would further facilitate this independence to ensure that traversal along each dimension means interpretable data generation. This seems to be controversial to analysis in Fig. 4, where different latent features show strong correlations to each other.

Is there an automatic/systematic way of inferring metric for better separation of populations?

4. Line 331 "All models were trained for 10 epochs on the NVIDIA P100 with 100GB of memory". Please justify 10 epochs. 100GB GPU mem is clearly incorrect.

Reviewer #2 (Remarks to the Author: Reproducibility):

The main idea behind the method is straightforward. However, the code/implementation cannot be evaluated without sufficient details.

Github link https://github.com/GelatinFrogs/ME-350VAE_Architecture is broken.

Reviewer #3 (Remarks to the Author: Overall significance):

Ternes et al. present multi-encoder variational autoencoder (ME-VAE) architecture for learning informative features from single-cell multi-channel image data. The goal is extremely significant in the field of bioimage analysis. Various approaches have been suggested during recent years to learn unbiased features instead of classical handcrafted features. These approaches enable more automated analysis solutions and importantly even robust models that can be applied to different datasets. The problem is still unsolved and the manuscript presents one possible solution. The

benefit of the ME-VAE architecture presented is that it does not need any labeled data to learn the features such as in supervised learning approaches. However, ME-VAE is dependent on the knowledge of the uninformative transformations present in the data so that these can be ruled out in different encoding blocks to extract biologically informative representation in single-cell image data. Some transformations, such as rotation, are obvious, but often the challenging transformation in the data is unknown. As an example, in large datasets, experimental batch effects cause many problems for representation learning tasks (and when using classical features as well), and typically cannot be well modelled. Thus, the significance to the field is lowered in the current version of ME-VAE methodology as the users need to know these uninformative transformations present in the data. These limitations are taken into account by the authors in the discussion. I still do think this is an interesting study and could lead to more practical solutions in the future. Authors also mention in discussion that "Future applications of this architecture will allow complex features such as texture, patterns, and distribution to be extracted from single cell images without the hassle of disentangling dominant uninteresting transform features", so maybe this problem is already being studied by them.

Reviewer #3 (Remarks to the Author: Impact):

In its current form, the most appropriate journal could be Communications Biology. Solving the limitations listed would improve the impact.

The manuscript presents one approach to tackle the problem of extracting biologically meaningful unbiased features. This is an important topic in the field of bioimage analysis, especially how to learn meaningful representation without annotated data. The manuscript presents one approach to solve the problem but does not introduce novel ways of thinking in the field.

Reviewer #3 (Remarks to the Author: Strength of the claims):

The major experiments missing from the manuscript are:

1. The authors compare their ME-VAE method to standard variational autoencoder and also to variational autoencoder with corrected output. These comparisons are important to show that ME-VAE performs better than simpler VAE approaches, however, the main question should be whether ME-VAE performs better than currently used approaches. Fig 2a) includes an example of comparing two features between two ligands. Later in Fig 2d) the authors present additional feature that is inferred from visually going through the data. This single feature presented in Fig 2d gives much better separation than standard VAE approach. These classical features should be compared to standard and multi-encoder VAE to see how well existing solutions enable separation of clusters.

2. As only two ligands are compared in results presented in Fig 2, I would expect the above mentioned classical feature comparison to be included also in Fig 3.

3. In addition to the point made in 2. regarding Figure 3, this experiment including all 6 ligands could benefit from quantitative measurements instead of only UMAP visualization. The data could be clustered and compared using some clustering performance metric. This evaluation would quantitatively show whether the ME-VAE improves currently available methods.

Here are minor corrections suggested to improve the quality of the manuscript:

- 4.

L38: immunofluorence -> immunofluorescence

L120: clusterizability and serperability: I am not sure if these are proper words

L170: nucleous -> nucleus

L176: unformative?

L331: NVIDIA P100 with 100GB memory? Did the P100 really had 100GB GPU memory or the computer had 100GB RAM?

Ref 17 is missing volume and issue information, Ref 18 is missing a title.

L581: recocnstruction -> reconstruction

Fig 2.d) The cluster purity pie charts could include labels (Cluster1 left? and Cluster2 right?)

Suppl. Fig. 3 title: "UMAP clusters" -> should replace clusters with visualization etc. as UMAP does not provide clusters, only dimensionality reduction.

L617: EFGR -> EGFR

Reviewer #3 (Remarks to the Author: Reproducibility):

1. The authors share the code to train ME-VAE model, however, they do not share models trained and used to produce results in the manuscript. Or at least I was not able to find these. In addition, their code includes only an example version of ME-VAE including two parallel encoding blocks and no image data generators to prepare data for these blocks. The authors make a point that these global uninformative features are data specific which is true, but it would make reproducibility much easier by including the encoding blocks and generators used in the manuscript as an example.

Author rebuttal, first version:

Comments from Reviewer #1

Summary

The paper titled "ME-VAE: Multi-Encoder Variational AutoEncoder for Controlling Multiple Transformational Features in Single Cell Image Analysis" by Luke Ternes and colleagues describes a novel computational model called Multi-encoder VAE (ME-VAE) for single cell image feature extraction that removes specified uninformative features by making them uniform and invariant across the reconstructions, using modified pairs of transformed input and output images by self-supervised transformation, and utilizing multiple encoding blocks. Using the ME-VAE to control for these multiple transformational features, the authors are able to extract biologically meaningful and transform-invariant single cell information and better separate heterogeneous cell types. The approach is novel, aims to address an important problem, and results in improved downstream results compared to the Standard VAE using no informed transformations. The authors also illustrate the ability of ME-VAE for multi-modal integration and comparison.

I do think this is an important paper but it needs major revisions (as I detail below) and seems more appropriate for *Nature Comp Sci* or *Comms Biology*. However, if the authors make the changes suggested and do a great job it could be appropriate for *Nature Comms*.

There are key limitations to this work, first the lack of details pertaining to generalizability and scalability, and the reduced clarity in presentation of the data, along with incomplete explanation of figures and equations. The manuscript feels rushed and not quite ready for submission, adding to the lack of clarity and readability.

Specific Comments

Comment 1.1, 1.2, 1.3:

- The first limitation is the lack of generalizability to other emerging multiplexed technologies such as CODEX, or MIBI.
- As mentioned in the introduction, there are upcoming multiplexed imaging technologies. In the current work, the authors only show ME-VAE on CYCIF data. For generalizability of such novel methods, it is essential to demonstrate ME-VAE on one other imaging technology. There is public data available for both CODEX and MIBI.

For example see:

- <https://portal.hubmapconsortium.org/docs/assays/codex>
- <https://www.angelolab.com/mibi-data>
 - In the last section of Results (A), the authors mention about generalizability and scalability. To address generalizability, please refer to comment #1

Response: 1.1, 1.2, 1.3:

- We would like to thank the reviewer for their suggestion for improvement and their recommendation of possible datasets. We conducted the additional experimental testing of Standard VAE and ME-VAE in a CODEX tissue dataset to demonstrate generalizability to other emerging multiplexed technologies. This included normalization, segmentation, tiling, and image processing to prepare single cell image tiles for encoding as well as the actual analysis with the proposed deep learning architectures. We have added a figure to the supplemental showing the ME-VAE's application compared to Standard VAE in the CODEX tissue dataset (Supp Figure 8). The improved clusterability compared to Standard VAE illustrates that the architecture has the ability to generalize to other multiplexed technologies. The other notable thing about the CODEX dataset is that it is tissue imaging data instead of the cell line data used in the main paper, further showing the architecture's ability to generalize for multiplexed tissue imaging dataset.

Comment 1.4:

- To address scalability, please show runtime benchmarks of ME-VAE against Standard VAE for one of the experiments (e.g. between Figure 1c-e)

Response 1.4:

- We acknowledge that we did not sufficiently describe scalability and we have addressed this in the revised text. Runtimes for all tested architectures including standard VAE, VAE with corrected output, additional deep learning models including β -VAE, Invariant C-VAE which reviewers suggested for comparison of downstream analysis have been added to Figure 1. The results of the time analysis shows that although the ME-VAE is slower than other architectures, the amount is negligible, which is expected since the architecture size isn't much bigger than the standard VAE model.

Comment 1.5:

- Regarding known controllable transformations: The results are shown for features that are known controllable transformations. These are then used as self-supervision to extract invariant features during model training. What about the case of noise-induced transformations that are unknown? Further, some of the known uninformative transformations such as rotation and polar orientation are not independent features. How do we know that these uninformative features are not getting mixed across encoders?

Response 1.5:

- We focus on identifying transform-invariant biologically meaningful features as we know that there are uninformative features that drive the observed difference between biologically similar images, skewing the results in undesired ways. As stated in the discussion, the limitation of the model is that the uninformative feature of interest and its transformation must be known. We show that our model is able to remove the uninformative features (as shown in Supp Figure 1). Potentially, if there are unknown noise-induced transformations, they will drive difference and skew the result so we could evaluate them and consider them as uninformative features by identifying transformations iteratively. For instance, rotation and polar orientation were first observed in Standard VAE so that we devise transformed images as a self-supervised signal to remove these uninformative features, which is our motivation and key contribution of the proposed work.
- There are likely countless artifacts and uninformative biological features that will crop up

throughout every researcher's experiments. Designing a model that can correct for all of them without prior knowledge would be a monumental if not impossible task, because as discussed, the model will not know what is of biological interest or not. As an example, for our MCF10A experiments, size/shape were not of biological interest, while for others they might be. With the proposed methods it is necessary that the transformations of these features are known, but it is an unfortunate truth that if something cannot be calculated, it can't be corrected.

- To answer non-orthogonal transformation such as rotation and polar orientation, we actually evaluate them in Supplemental Figure 1. We observed that controlling for one feature does not significantly impact the other dominant transformation features (i.e. polar orientation). The VAE with transformed output is shown to work on simple transforms such as rotation, but pairs of complex transformations like rotation combined with polar orientation prove too difficult. Both the β -VAE and invariant C-VAE also show strong correlations as well between the uninformative features we wanted to ignore and the latent space (Fig. 1 e/f and Supplemental Figure 1d/e). Finally, when both uninformative features are controlled for using the proposed ME-VAE with transformed image pairs, we see a decorrelation in both uninformative features, indicating that the VAE reconstructions learned to overcome them and focus on underlying features that better separate cell populations. As the proposed architecture allows for image pairs (i.e., transformed image and the original image) to be randomly and each feature is retained in parallel encoders in a self-supervised fashion, artifacts within a polarity correction encoder will not be present in a rotation correction encoder and vice versa.

Comment 1.6:

- Size and shape of a cell are important and informative features. For example, depending on the tissue being imaged and the context, certain cell types are larger than others (e.g. macrophages), or they might have a certain shape (spindle-like). This information is essential to be able to segregate them. Is it then justifiable to convert these features to being uniform and invariant across transformations?

Response 1.6:

- It is justifiable depending on the context of the analysis being performed. In the cell line MCF10A data we were analyzing, shape was not a feature of interest. Moreover, size is an easily extractable shape feature that can be captured during cell segmentation and added back into analysis later if desired. To address this point, when analyzing the additional CODEX dataset, we did not control for size and shape (only controlling for polar orientation and rotation). We see good performance of identifying cell phenotypes and are able to extract macrophage population (see supplemental figure 8 with high CD68 expression in single cluster). The decision to include or exclude uninformative/informative features such as size/shape control is for the justification of the user of the ME-VAE to their specific task and biological questions. The ME-VAE just enables them to make that choice for a general use case.

Comment 1.7:

- The crux of this work relies on transformed image pairs. What are these image pairs – an input image and its transformed output? Or are these the two transformed images, one for rotation and one for polar orientation?

Response 1.7:

- The proposed ME-VAE takes in a number of transformed input images equal to the number of features you are looking to control for. The input to each encoder is transformed such that one of the features (rotation/polar orientation/other feature) is randomly changed across all images. The output serves as the starting point for all random transformations. It is

transformed such that the features are controlled for prior to random alteration. By doing this, we extract meaningful latent representation by removing uninformative features in a self-supervised way (i.e., we do not need to teach our deep learning model to correct these uninformative features). As we use these known transformations (i.e., rotation and polar) as self-supervised signal, ME-VAE can remove these features and extract biologically meaningful features.

Comment 1.8:

- **Figure 2:** Legend says 'Rotation angle of cells are shown in UMAP embedding to show the influence of unimportant features on downstream analysis'. Where is this shown in the figure?

Response 1.8:

- Figure 2 shows the regional cell images throughout the UMAP space, and it can be observed that cells within the UMAP space are organized by their rotation angle. Below the regional cell images, we've also included a plot of the UMAP space colored by the rotation angle to illustrate how the effect governs the UMAP embedding.

Comment 1.9:

- **Figure 2b:** What is the input to k-means? Also mention what each dot is in the UMAP or k-means plot. How many dots are shown in the figure?

Response 1.9:

- As described in the methods of the paper, k-means clustering was performed on the VAE encoding spaces, so the input to k-means was the encodings for each cell. Each dot in UMAP and k-means plot is a single cell. The number of dots shown in the figure are the number of cells used in the experiment (Sample size of 15,898 for 2 ligand dataset in Figure 2 and 73,134 for 6 ligand dataset as described in Figure 3).

Comment 1.10

- What are regional cell images (e.g. in **Figure 2b-c**)? The blue square seems to have many dots whereas the zoomed in regional cell image shows 25 cells. Please also provide one higher resolution color image, with an explanation of biologically relevant features (stain localization, intensity, and subcellular pattern) within this zoomed-in regional cell image

Response 1.10:

- We acknowledge that we did not sufficiently describe this. Regional cell images are representative images of cells sampled from various regions of the umap embedding space to give visual context to the features that are being separated. The region that is being sampled is the blue square, which does contain many cells; however, for display, cells are snapped to a grid of 5x5 (with many cells inhabiting the same space) and then one representative cell is shown for each point in that grid. More information about the regional umap visualization can be found here: <https://doi.org/10.1117/12.2512660>

Comment 1.11:

- What are the radial slopes for **Figure 2c**? Since this is computed by fitting a regression line, how can a same/similar slope distinguish similar distributions for different cell types?

Response 1.11:

- We believe that the reviewer referred Figure 2d in the original draft (in the revised version Figure 2g). Looking at the clusters and cell types extracted from the ME-VAE, we qualitatively inferred that a potentially distinguishing feature between the two populations is the radial distribution of the stain. So, we handcrafted a feature that was not obvious from prior knowledge to attempt to capture what the ME-VAE was encoding. To do this, we calculated the slope of the distribution (see Methods C and Supplemental Figure 2) using the

mean intensity at each radial distance and taking the slope. This metric tells us the general trend of the stain, i.e. where it is located in the cell and how it changes as it goes from the nucleus to the membrane. PBS has a higher radial slope, which indicates that the intensity of the stain increases more rapidly toward the membrane, while TGFβ has a lower radial slope, indicating that it decreases toward the membrane. This can be re-affirmed by looking at the sampled images of each populations of figure 2f (previously 2c). The cells show distinct localization at different radial locations within the cell. Note that this subcellular feature was not captured by simple mean intensity profile and thus we do not observe separation between PBS and TGFβ (Figure 2a).

Comment 1.12:

- **Figure 2c:** The cluster purities from radial slope metrics, however, are still lower than the full ME-VAE cluster purity, indicating more features beyond the radial slope are being extracted from ME-VAE: Is this really a case of more features or is this a case of ME-VAE being overfit to the 'noise' that got extracted?

Response 1.12:

- Classification of labels was not included in the loss function of the models; therefore, overfitting would not work to separate the ligand conditions. Improvement in classification indicates the presence of learned features which the model captured agnostic to the ground truth labels of the data. Also, we demonstrated that ME-VAE results inform us hand-crafting a new metric to separate two populations better (i.e., initial naïve metrics vs inferred radial slope metrics). This confirms and validates the ME-VAE's encodings in real world biology and demonstrates that more than just noise is being extracted.

Comment 1.13:

- **Figure 3:** 'Size does show some distribution in the UMAP': Please highlight this in **Figure 3, Supplementary 3b**

Response 1.13:

- Size intensity profile/distribution in UMAP space is prominently shown in figure 3 (being the second listed in the subplots). The size distribution can already be observed in Supplemental Figure 3 from the sampled cell images from varying places in UMAP space.

Comment 1.14:

- In **Figure 4a** (bottom), each column is a cluster and is identified by a set of differentially expressed markers. Why is then each row showing a different set of differentially expressed markers per column? Same comment for **Supplementary Figure 4**

Response 1.14:

- With visualization, we are limited to showing a few markers in a single image of a cell. To capture as much variation, we showed the same cell with different sets of three markers in each row. The columns are the representative cell for each cluster. The choice of markers is the same across the columns, but the differences in expression illustrate that each cluster is capturing differences in expression.

Comment 1.15:

- Please give an example of 'morpho-spatial profiles' (mentioned in Results D)

Response 1.15:

- An example of this is given earlier in the same paragraph where we describe ratios of cell and nuclear size. Some of the aggregated features exhibit a larger relative nuclear to cell size ratio, whereas others exhibit small nuclear to cell size ratio. Also, in Figure 4 b), we reported that ME-VAE aggregated features extracted subcellular or compartmental intensity profiles to demonstrate an example of 'morpho-spatial profiles'.

Comment 1.16:

- **Supplemental Figure 4b:** Please highlight or mention in the legend the row/column number where the following is observed: a 'single aggregated feature that shows significant correlations shows correlates to every RPPA pathway activity profile (Supplemental Figure 4b). Second, there is a single RPPA pathway that correlates to every standard VAE aggregated feature.'

Response 1.16:

- We would like to thank the reviewer for this suggestion to improve the interpretability of our figure. We have pointed this out in the figure description, referring directly to the relevant rows and columns.

Comment 1.17:

- In Results D, please add citations for 'known biology', 'known literature'.

Response 1.17:

- All examples of known biology and known literature in this section are followed with a citation (see Stat3, cyclinD1, and p21 examples in text). The only one that does not is where we state DAPI is correlated to DNA pathway. Since DAPI is a marker for DNA, we found this correlation self-explanatory. We have rephrased this sentence to better illustrate what we were attempting to convey.

Comment 1.18:

- In the Discussion, there is mention of 'augmenting' the model. What would an example for an augmented feature be and how would this be transformed for the ME-VAE

Response 1.18:

- What we are describing here is not augmenting a cell feature. What we are describing is augmentations that can be added to the model itself in future iterations of the ME-VAE to improve its efficacy with new developments in the computer vision community. In the text we give the example of augmenting it with a discriminator, which is an adversarial network occasionally added onto the standard VAE used to improve reconstruction quality. The point we are trying to make here is that the concept of the ME-VAE is versatile not only is it easily amenable to future improvements with new discoveries such as novel loss functions, but it can also be worked into new or existing deep learning architectures fairly easily. Since it is a simple to implement concept using existing pieces architecture blocks, other forms of Deep Learning can implement the concept into their architectures as well.

Comment 1.19:

- How reliable was the EGFR channel for segmentation? For cells where the EGFR signal is not clear, would it not help to identify such cells by using additional nuclear markers for segmentation? For the extended dataset, was the segmentation again done using only the EGFR channel? If only EGFR was used, why was this the case?

Response 1.19:

- Segmentation was done simultaneously for the entire dataset, and it was done only once. For whole cell segmentation, CellPose utilizes two channels (one of which is always nuclear), so the segmentation used EGFR and DAPI. The manuscript has been updated to better convey this. EGFR was chosen because it produced the highest quality segmentations of the manually annotated test segmentation set. This data and justification were not included as the purpose of this paper is not in the segmentation approach. For an additional experiment with CODEX dataset, Mesmer segmentation pipeline on the Hoechst and CD71 markers because this qualitatively produced the best results.

Comment 1.20:

- **Figure 5:** 'ME-VAE features used for comparison were the features with largest correlation to the respective CYCIF marker'. Why not compare CYCIF with the ME-VAE clustered (aggregate) features? The authors already point out that they do hierarchical clustering on the ME-VAE feature 'to reduce the feature dimensionality and reduce spurious correlations in the biological findings. This comparison would also give an idea of how the clustered features look like.

Response 1.20:

- We acknowledge that we did not fully support this claim. To address this concern, a similar figure to figure 5 but using the Aggregated features has been added to the Supplemental Figure 8. We are not as worried about spurious correlations in the figure since the purpose is to show larger statistical separation of cell populations and not the correlation to CYCIF markers. Looking at the aggregated features, we do see slightly less significance compared to the single features (which is expected since aggregated features will average out some of the signal), but the overall separations are still shown to be greater consistently than the CYCIF markers.

Comment 1.21:

- **Figure 5:** Further, how many ME-VAE and Standard VAE features were there? Is there any close correspondence between the z-scores in either column per row?

Response 1.21

- There were 512 features for each; however, Standard VAE is not shown in figure 5, only the ME-VAE. Due to other comments a new figure showing the same results with the set of 10 ME-VAE aggregated features (Supplemental Figure 7). The features that were selected were those with the largest single cell correlation in z-score expression between CYCIF intensity and VAE Encoding.

Comment 1.22:

- ME-VAE encoding features were restricted to 18 single features for each'. Does this mean that 1 ME-VAE feature = 18 single features? If this is the case, how were 18 single features assigned to one ME-VAE feature?

Response 1.22

- No. For more interpretable and visualizable analysis, the 512 features were restricted to 18 chosen features for each analysis. As detailed in the methods, the features were chosen to optimize a cluster variability function (also shown in methods).

Comment 1.23:

- Equations in Methods B: Please explain all the variables and what the equations do.

Response 1.23:

- We acknowledge that we did not sufficiently describe this. The loss equations for each VAE architecture have been re-written and all variables are properly defined. As described in the methods, these equations are the loss function used for training the models (it is the function the model is trying to optimize). For reproducibility, we have also added the code with precise implementation, further details describing variables and equations, as well as shared trained models with parameters in Github.

Comment 1.24:

- **Figure 1:** Mention the data used, number of cells etc. in the Figure legend

Response 1.24:

- The relevant information has been added to the figure legend.

Comment 1.25:

- What are the data dimensions for the RPPA dataset?

Response 1.25:

- The RPPA dataset consisted of 295 original protein markers which were aggregated to 9 key pathways. The dataset was done in bulk on 6 ligand populations with 3 replicates for each, totaling 18 datapoints. (Exact plotting and analysis of these datapoints is detailed and shown in supplemental figure 10).

Comment 1.26:

- There are two cell numbers reported – 71314 and 73,134. Is the former after pre-processing the images?

Response 1.26:

- We apologize for the typo error. There are 6 instances of 73,134 being used and only one instance of 71314 be used. This was a typo and has been fixed.

Comment 1.27:

- **C. Evaluation metrics:** Explicitly state how the slope was calculated: was it using the β from the regression equation?

Response 1.27:

- We apologize that this was left unclear and thank the reviewer for the opportunity to clarify. As is described in Methods C and Supplemental Figure 2, the scipy linregress terminology for function outputs is slope and intercept. Here we used the slope as the metric and ignored the intercept. We have clarified this in the revised manuscript.

Comment 1.28:

- Which clustering method was used from the seaborn clustermap function?

Response 1.28:

- We apologize that this was left unclear and thank the reviewer for the opportunity to clarify. The default hierarchical clustering method (Euclidean) was used. This has been clarified in the revised manuscript.

Comment 1.29:

- Please spell check the document. There are typographical errors relating to words e.g. decrease, separability, reconstruction, hierarchical, python, spearman, as well as word repeats.

Response 1.29:

- We apologize for typographical errors. We have addressed this in the revised text.

Comment 1.30:

- **Supplemental Fig 4:** correct the text to reflect Standard VAE.

Response 1.30:

- This has been fixed in the revised text.

Comment 1.31:

- **Figure 5:** Specify which type of ANOVA was used, and what was the p-value or F-statistic and depict this in a figure.

Response 1.31:

- The Anova was performed through python statsmodels package using type 2 anova calculating the F-statistic. This has been clarified in the text. We have added Anova F-statistic scores to the figure.

Comments from Reviewer #2

Summary

Ternes et al. propose an extension of the classical VAE (variational autoencoder) for single cell image analysis for the purpose to extract biologically more meaningful latent representation of the input images. The main motivation is that the vanilla VAE tends to identify non-biological images features present in the dataset, such as rotation, scale etc, which can be viewed as confounding factors/ biases in the training dataset. The authors propose a method, called ME-VAE, to remove these non-informative features from the latent representation, hoping that the resulting new latent representation can lead to a better clustering or characterization of cell types/states.

The main idea behind ME-VAE is data normalization plus data augmentation. It generates a new set of target images that have been properly normalized, corrected based on a given set of predefined transformations. It then trains the model with random transformations of the input images, forcing the model to learn to ignore these transformations and focus on biological more meaningful features. The authors demonstrated that ME-VAE was able to yield biologically more meaningful representations than VAE through clustering and correlation analysis.

Specific Comments

Comment 2.1:

- It seems to me a more focused, specialized journal is more appropriate for this manuscript.

Response 2.1

- We propose a novel deep learning approach for single cell image analysis and demonstrate that the proposed method improves analysis by making distinct cell populations more separable compared to traditional and current VAE architectures. We also observe that the proposed method improves correlation to other analytic modalities by enhancing phenotypic differences between cells. As there are many naïve applications of standard VAE in bioimaging dataset and single cell multiplexed imaging analysis currently relies solely on mean intensity profiles, we believe that our manuscript of significant interest is appropriate and timely important topic for this journal.

Major Comments

Comment 2.2:

- The authors focus on comparing ME-VAE to vanilla VAE. However, this is highly biased for several reasons. First, there are several other recent works on single cell image analysis that have not been properly discussed, and certainly not experimentally compared. I highly recommend the authors take a close look at the methods described in the following paper and carry out a thorough comparison analysis against these existing methods.
 - MCMICRO: A scalable, modular image-processing pipeline for multiplexed tissue imaging by Schapiro et al.

Response 2.2:

- We have added several comparison methods to our analysis (see updated Figure 2, Results A/B and Methods A/C). Included among these are two modified VAE architectures and more classically extracted features. MCMICRO allows modular and sequential steps to perform analysis on multi-channel images, but it generates image segmentation and cellular features (mainly mean pixel intensity, i.e., protein expression) primarily. Note that the proposed approach in this paper focuses on identifying suitable representations that capture complex imaging features instead of using classical cellular features. We also compare classical cellular features one could extract from MCMICRO pipeline with VAE features in both Figure

2 and Figure 5 by demonstrating separability of ligands condition.

Comment 2.3, 2.8:

- Second, going back to the VAE method itself, it is well known that VAE does not handle confounding factors well. There are many existing works on how to correct confounding factors on VAE. Some of these methods have also been proposed for single cell genomic data analysis. A few references include:
 - Moyer, D. et al. (2018) Invariant representations without adversarial training. Advances in Neural Information Processing Systems, 31, 9084–9093.
(github: <https://github.com/dcmoyer/invariance-tutorial/blob/master/tutorial.ipynb>)
 - Deep Generative Modeling for Single-cell Transcriptomics, Romain Lopez et al, NatureAlthough they are applied to different types of datasets, the methods themselves can be applied to single cell image analysis as well. Instead of comparing with vanilla VAE, the author should compare with these more recent extensions of VAEs.
- Regarding the method itself, the authors should compare with the vanilla VAE using normalized/corrected images, that is, applying VAEs on normalized images instead of raw images.

Response 2.3, 2.8:

- We have received many comments asking for comparisons to many different methods from each author. This response seeks to address all those comments at once since they are asking the same thing. From the reviewers' recommendations and from the list of other architectures we describe among the prior works, we have selected a few additional comparisons to add to our analysis. We adapted and implemented them to fit the design constraints of this experiment, including matching the architecture depth/size. The comparison is done in Figure 1 and 2 to perform the task of removing uninformative features from the encoding space and of separating 2 ligand populations with a single channel image.
- We see that most do not perform much better than a Standard VAE in this context, and although the invariant C-VAE by Moyer, D. et al. recommended by the reviewer performs somewhat better, it does not perform as well as the ME-VAE. This demonstrates that the proposed method outperformed the existing methods reported in references and suggested by the reviewers. For subsequent analysis beyond this figure, we decided it was unnecessary to perform the other methods in the larger dataset since they already performed worse than the ME-VAE in the restricted dataset. In total, the ME-VAE is now compared to 5 other methods/arrangements of VAE and is tested in 3 datasets (MCF10A with 2 ligand/1 channel, MCF10A with 6 ligand/23 channel, and CODEX tissue image with 20 channels to demonstrate generalizability of the proposed method by another reviewer).

Comment 2.4:

- It's also unclear to me why VAE is a good method for single cell image analysis. VAE is a generative model. The Gaussian prior applied on the latent variable tends to pull all representations toward the origin, and consequently reduces the separation between different cell types. The authors should provide a justification on why VAE is a good model for single cell analysis, and why it is better than a simpler denoise auto-encoder, the non-generative model.

Response 2.4:

- Many recent studies demonstrate that VAE approaches produced encouraging results by finding previously unappreciated cellular structures, allowing for accurate predictions across a variety of areas and outperformed classical methods for analyses of hand-crafted features (<https://doi.org/10.1101/227645>). The main difference between autoencoders and variational autoencoders is that the latter impose a prior on the latent space. A VAE is an

autoencoder whose encodings distribution is regularized during the training in order to ensure that its latent space has good properties allowing us to generate some new data. Moreover, the term “variational” comes from the close relation there is between the regularization and the variational inference method in statistics. In fact, the high degree of freedom of the autoencoder that makes possible to encode and decode with no information loss (despite the low dimensionality of the latent space) leads to a severe overfitting implying that some points of the latent space will give meaningless content once decoded. A VAE can be defined as being an autoencoder whose training is regularized to avoid overfitting and ensure that the latent space has good properties that enable generative process.

Comment 2.5:

- The approach works for pre-defined, well-known confounding factors such as rotation, scale. But what about latent features not associated with a well-defined transformation? It is well known that deep learning models tend to pick up correlated features that are not biologically meaningful. How do you plan to handle these features, which are a) not known beforehand, and b) may not be associated with a rigid simple transformation

Response 2.5:

- As laid out in the discussion, these are the limitations of the proposed method. From our experience, using VAEs can be an iterative process, and often time the failure of the first attempt will illustrate which features in the dataset present confounding factors. After initial rounds of training and observation, we found features such as rotation were skewing results, followed by orientation and size features on subsequent iterations. This can be done if the features are expected prior to training, but likely for most researchers, the process will require exploration of their dataset. Simple and rigid are not necessarily prerequisites for the transformation. As described in the methods, in order to correct shape, we performed a non-rigid registration of the cell to a circle. Complex shape is not a feature that is easily described, but none-the-less can be overcome using a non-rigid transformation to a set standard. There are certainly complex features that go beyond the limitations of what the ME-VAE can handle and we are in no way selling this as a method that can fix any confounding feature. The ME-VAE, however, does allow a method that outperform many of the other VAE methods and is able to correct for several factors simultaneously. Moreover, even the current comparable methods come with similar limitations. The invariant C-VAE that was recommended for comparison still requires quantified/classified information to be input into the model and suffers from the limitation of being able to quantify the feature (such as shape) which might be a non-trivial task.

Comment 2.6:

- Because the current model doesn't address batch effect, the better clustering shown in Figure 2 can potentially be associated with the batch effect. I would recommend testing the model on biological replicates of the same cell types to show that cells of the same type from different batches are mixed.

Response 2.6

- We would like to thank the reviewer for pointing this out so that we can better clarify how this is addressed in our dataset. In fact, the dataset was comprised of 3 replicates of each ligand on different plates, wherein each plate had an additional 3 replicates of each ligand in different wells, and each well had 9 different fields of view taken. The manuscript has been updated to include this information. The fact that the ligands cluster well with themselves despite the number of replicates indicates that the clustering is not due to simple batch effects.

Comment 2.7:

- I would also like to see the results from the samples not in the training dataset. If the features are truly biologically meaningful, I would expect to see similar results on these samples as well.

Response 2.7:

- In a typical supervised learning setting, it makes sense to test model with unseen dataset. However, we are not training our ME-VAE model in a supervised setting and we demonstrate performance of unsupervised clustering result using extracted latent space with known labeled information. The results for clustering, umap distributions, regional cells, integration, and separability are all taken from using the entirety of the dataset. We see cells clustering based on the phenotype and ligand across the entire dataset and see no population of isolated cell images that would convey an overfitting of the model. Also, to demonstrate the features are truly biologically meaningful, we demonstrate correlations to other analytic modalities (Reverse Phase Protein Arrays pathway activity) as validation.

Comment 2.9:

- Please use standard metrics such as ARI, NMI to evaluate clustering qualities.

Response 2.9:

- We would like to thank the reviewer for their suggestion of supplemental clustering metrics. We have added NMI to the relevant quantifications and figures.

Comment 2.10:

- I also highly recommend the authors to test the method on a separate, ideally public dataset.

Response 2.10:

- We would like to thank the reviewer for their suggestion for improvement. We have added a figure to the supplemental showing the ME-VAE's application in a publicly available CODEX tissue dataset to demonstrate generalizability to other emerging multiplexed technologies (Supp Figure 8). The improved clusterability compared to Standard VAE illustrates the architecture's ability to generalize to other multiplexed technologies. The other notable thing about the CODEX dataset is that it is tissue data instead of the cell line data used in the main paper, further showing the architecture's ability to generalize for multiplexed tissue imaging dataset.

Minor Comments

Comment 2.11, 2.12:

- The description of VAE models in Method B should be substantially improved. Notations are non-standard. Variables are often not defined or not referenced. Equations are unlabeled, and which loss function is for which model is not mentioned.
 - o Equation of L_e seems to use terms T_i^{-1} .
- The ELBO of VAE contains a reconstruction term and a KL-divergence term encouraging smoothness of the latent space. The KL term seems to be missing from the loss functions.

Response 2.11, 2.12:

- We apologize that this was left unclear. The equations describing the loss functions of the VAE architectures have been re-written in the revised manuscript.

Comment 2.13:

- Since the vanilla VAE uses isotropic Multivariate Gaussian for prior, the KL term will facilitate different dimensions of latent z to be independent with each other. Later proposed disentanglement methods would further facilitate this independence to ensure that

traversal along each dimension means interpretable data generation. This seems to be controversial to analysis in Fig. 4, where different latent features show strong correlations to each other.

Is there an automatic/systematic way of inferring metric for better separation of populations?

Response 2.13:

- We would like to thank the reviewer for bringing up this question. As the reviewer pointed out, VAEs assume each data point is i.i.d. generated, which means we do not consider any correlations between the data points. Due to the i.i.d. assumption, VAEs only optimize the singleton variational distributions and often fail to account for the correlations between data points, which might be crucial for learning latent representations from datasets where a priori we know correlations exist. For instance, we know marker-wise correlation in our multiplexed imaging dataset and there exists correlation between data points across different ligand treatments. In fact, we observed standard VAE shows less correlation across features but fails to extract biological meaningful information as shown in Supplemental Figure 4. On the other hand, when learning latent representations with the proposed approach, ME-VAE shows correlation structure across latent representation (Figure 4) and feature aggregations show high correlation to almost all RPPA pathway compared to standard VAE features.
- In addition, although each dimension means interpretable data generation, it is almost impossible to interpret 512 latent space as meaningful biological features so we simply consider dimension reduction by aggregating correlated features into low dimension space and correlate with RPPA pathway activity. Although standard VAEs push for independence, many architectures have been developed attempting to address the fact that this does not guarantee interpretability or non-noisy content being encoded. Even among these, there is no agreed upon standard and in many applications we see only minutely better performance (<https://doi.org/10.1101/2021.09.02.458673>), and interpretability is even with further methods such as latent space arithmetic is limited. Our own results of these methods re-iterate this, and many of these methods come with limitations of their own, including extensive hyperparameter tuning. Specifically, we observed a tradeoff between the Standard VAE's ability to reconstruct samples and disentangle features, as indicated by the inverse relation of reconstruction and latent space correlation in β -VAE compared to Standard VAE. We use standard metrics such as NMI, ANOVA and the post-hoc pairwise Tukey p-test to measure separation of populations.

[There is no Comment 2.14 on the assessment, it skips from 2.13 to 2.15]

13	facilitate this independence to ensure that traversal along each dimension means interpretable data generation. This seems to be controversial to analysis in Fig. 4, where different latent features show strong correlations to each other.
	Is there an automatic/systematic way of inferring metric for better separation of populations?
15	Line 331: "All models were trained for 10 epochs on the NVIDIA P100 with 100GB of memory". Please justify 10 epochs. 100GB GPU mem is clearly incorrect.

Comment 2.15:

- **Line 331:** "All models were trained for 10 epochs on the NVIDIA P100 with 100GB of memory".

Please justify 10 epochs. 100GB GPU mem is clearly incorrect.

Response 2.15:

- The 10 epochs was chosen because after 10 was the amount of training needed before there

was an observed plateau in loss. The number of epochs was kept consistent for accurate comparison. This has been better explained in the manuscript. The exact resources used for training are an NVIDIA P100 GPU with 100GB of disc space and 100GB of RAM. This has been better specified in the revised manuscript.

Comment 2.16:

- The main idea behind the method is straightforward. However, the code/implementation cannot be evaluated without sufficient details.

Github link https://github.com/GelatinFrogs/ME-350VAE_Architecture is broken.

Response 2.16:

- The link is not broken. From the link you provided, the "350" is not included in the paper. I am going to assume the "350" is the line number of the pdf or whatever document you are working in accidentally getting copied. Github link: https://github.com/GelatinFrogs/ME-VAE_Architecture

Comments from Reviewer #3

Summary

Ternes et al. present multi-encoder variational autoencoder (ME-VAE) architecture for learning informative features from single-cell multi-channel image data. The goal is extremely significant in the field of bioimage analysis. Various approaches have been suggested during recent years to learn unbiased features instead of classical handcrafted features. These approaches enable more automated analysis solutions and importantly even robust models that can be applied to different datasets. The problem is still unsolved and the manuscript presents one possible solution. The benefit of the ME-VAE architecture presented is that it does not need any labeled data to learn the features such as in supervised learning approaches. However, ME-VAE is dependent on the knowledge of the uninformative transformations present in the data so that these can be ruled out in different encoding blocks to extract biologically informative representation in single-cell image data. Some transformations, such as rotation, are obvious, but often the challenging transformation in the data is unknown. As an example, in large datasets, experimental batch effects cause many problems for representation learning tasks (and when using classical features as well), and typically cannot be well modelled.

Thus, the significance to the field is lowered in the current version of ME-VAE methodology as the users need to know these uninformative transformations present in the data. These limitations are taken into account by the authors in the discussion. I still do think this is an interesting study and could lead to more practical solutions in the future. Authors also mention in discussion that "Future applications of this architecture will allow complex features such as texture, patterns, and distribution to be extracted from single cell images without the hassle of disentangling dominant uninteresting transform features", so maybe this problem is already being studied by them.

In its current form, the most appropriate journal could be *Communications Biology*. Solving the limitations listed would improve the impact. The manuscript presents one approach to tackle the problem of extracting biologically meaningful unbiased features. This is an important topic in the field of bioimage analysis, especially how to learn meaningful representation without annotated data. The manuscript presents one approach to solve the problem but does not introduce novel ways of thinking in the field.

Response: Many of the recent extensions of the VAE that seek to improve the interpretability of the latent space simply modify the loss function used during training to encourage a result instead of forcing it. Two examples of recent architectures that use modifications to the objective function are

the β -VAE²¹ and the invariant C-VAE²², which attempt to apply pressure to model such that it will prioritize a more regularized encoding space and be more interpretable and invariable to specific features.

Unlike these previous attempts, the ME-VAE changes the actual deep learning architecture by adding multiple encoding blocks each for the purpose of removing a specific feature in a self-supervised setting, which we observe to have an increased performance. By doing this, we extract meaningful latent representation by removing uninformative features in a self-supervised way (i.e., we do not need to teach our deep learning model to correct these uninformative features). As we use these known transformations (i.e., rotation and polar) as self-supervised signal, ME-VAE can remove these features and extract biologically meaningful features. Thus we believe that we introduce novel ways of thinking in the fields.

Major Comments

Comment 3.1, 3.2:

- The authors compare their ME-VAE method to standard variational autoencoder and also to variational autoencoder with corrected output. These comparisons are important to show that ME-VAE performs better than simpler VAE approaches, however, the main question should be whether ME-VAE performs better than currently used approaches. Fig 2a) includes an example of comparing two features between two ligands. Later in Fig 2d) the authors present additional feature that is inferred from visually going through the data. This single feature presented in Fig 2d gives much better separation than standard VAE approach. These classical features should be compared to standard and multi-encoder VAE to see how well existing solutions enable separation of clusters.
- As only two ligands are compared in results presented in Fig 2, I would expect the above mentioned classical feature comparison to be included also in Fig 3.

Response 3.1, 3.2:

- We have received many comments asking for comparisons to many different methods from each author. This response seeks to address all those comments at once since they are asking the same thing. From the reviewers' recommendations and from the list of other architectures we describe among the prior works, we have selected a few additional comparisons to add to our analysis. We adapted and implemented them to fit the design constraints of this experiment, including matching the architecture depth/size. The comparison is done in Figure 1 and 2 to perform the task of removing uninformative features from the encoding space and of separating 2 ligand populations with a single channel image. We see that most do not perform much better than a Standard VAE in this context, and although the invariant C-VAE by Moyer, D. et al. recommended by the reviewer performs somewhat better, it does not perform as well as the ME-VAE. This demonstrates that the proposed method outperformed the existing methods reported in references and suggested by the reviewers. For subsequent analysis beyond this figure, we decided it was unnecessary to perform the other methods in the larger dataset since they already performed worse than the ME-VAE in the restricted dataset. In total, the ME-VAE is now compared to 5 other methods/arrangements of VAE and is tested in 3 datasets (MCF10A with 2 ligand/1 channel, MCF10A with 6 ligand/23 channel, and CODEX tissue with 20 channels.)

Comment 3.3:

- In addition to the point made in 2. regarding Figure 3, this experiment including all 6 ligands could benefit from quantitative measurements instead of only UMAP visualization. The data could be clustered and compared using some clustering performance metric. This evaluation would quantitatively show whether the ME-VAE improves currently available methods.

Response 3.3:

- We would like to thank the reviewer for the suggestion on how we can better describe the improved performance. The same clustering performance metrics that were performed in the restricted dataset (cluster purity and NMI) have been added to this figure as well. The quantitative measurements confirm what is shown in the UMAP visualization, that the ME-VAE performs better than the Standard VAE.

Minor Comments

Comment 3.4, 3.6, 3.7, 3.9, 3.10, 3.13:

- **Line 38:** immunofluorence -> immunofluorescence
- **Line 170:** nucleous -> nucleus
- **Line 176:** unformative?
- Ref 17 is missing volume and issue information, Ref 18 is missing a title.
- **Line 581:** reconstruction -> reconstruction
- **Line 617:** EFGR -> EGFR

Response 3.4, 3.6, 3.7, 3.9, 3.10, 3.13:

- We would like to thank the reviewer for taking the time to point out some of the typographic mistakes in the manuscript. These errors have been fixed.

Comment 3.5:

- **Line 120:** clusterizability and serperability: I am not sure if these are proper words

Response 3.5:

- We apologize for any typographical errors. "clusterability" and "separability" are the proper terms; however, both were spelled incorrectly. This has been fixed in the manuscript.

Comment 3.8:

- **Line 331:** NVIDIA P100 with 100GB memory? Did the P100 really had 100GB GPU memory or the computer had 100GB RAM?

Response 3.8:

- The exact resources used for training are an NVIDIA P100 GPU with 100GB of disc space and 100GB of RAM. This has been better specified in the revised manuscript.

Comment 3.11:

- **Fig 2d:** The cluster purity pie charts could include labels (Cluster1 left? and Cluster2 right?)

Response 3.11:

- We have added labels to the figures (now figure 2g) to improve interpretability, and have added a legend labeling the ligands to that subfigure as well to improve understanding of the cluster pie charts.

Comment 3.12

- **Suppl. Fig. 3 title:** "UMAP clusters" -> should replace clusters with visualization etc. as UMAP does not provide clusters, only dimensionality reduction.

Response 3.12:

- We agree. This was improper and confusing terminology. This has been fixed.

Comment 3.14:

- **Regarding Reproducibility:** The authors share the code to train ME-VAE model, however, they do not share models trained and used to produce results in the manuscript. Or at least I was not able to find these. In addition, their code includes only an example version of ME-VAE including two parallel encoding blocks and no image data generators to prepare data for

these blocks. The authors make a point that these global uninformative features are data specific which is true, but it would make reproducibility much easier by including the encoding blocks and generators used in the manuscript as an example.

Response 3.14:

- We apologize that we did not sufficiently share our models. We have addressed this and shared trained models for the model comparison with parameters in our Github, including a small sample images that go with them. We did not have the original models that were presented, so we re-trained and re-conducted the experiments for the comparison, and have uploaded the newest versions of the models. The results of the newly trained models performed similarly to the first and convey the same improvement compared to alternative methods. For this experiment we did not develop generators that can be shared for this implementation, nor do we claim in the paper to use them. For the sake of easy experimentation on our part, we pre-generated all the epochs' worth of augmented image pairs so we could try different architectures with more ease.

Reviewer comments, second version:

Reviewer #1 (Remarks to the Author: Overall significance):

The authors present ME-VAE, a DL architecture that removes known uninformative features by making them uniform and invariant across reconstructions, to improve downstream analysis of single cell imaging data. Using CYCIF images from MCF10A cells containing various biomarkers, the authors demonstrate that ME-VAE can clearly separate a TGF β +EGF population from controls, and outperforms standard VAE. The authors also show that there is a clear pattern of self-correlations between ME-VAE features, and identify representative clusters from these.

In the updated manuscript, the authors have incorporated changes to reflect most of the suggestions provided by the reviewers, especially the benchmarking to CODEX and other tools, and clarify comments for better understanding.

There still exists lack of clarity in the text and there are additional typos, wrong references introduced in the text which makes the rebuttal look quite unprofessional.

Below we state our comments for further clarification.

Reviewer #1 (Remarks to the Author: Impact):

Revision required. Manuscript is more suitable for Communications Biology

Reviewer #1 (Remarks to the Author: Strength of the claims):

In Response 1.5, authors explain that 'With the proposed methods it is necessary that the transformations of these features are known, but it is an unfortunate truth that if something cannot be calculated, it can't be corrected'. I would defer from this argument in that if something cannot be calculated, then attempts should be made to infer these so that they can be corrected. This is very true in biology as it is a mixed bag of many unknown and confounding features with very little that is known and the authors agree to this by stating the presence of 'countless artifacts and uninformative biological features'. If a model heavily relies on only the limited set of known features then this severely handicaps the model to generate new biological discoveries.

Is Response 1.10 added to the main text? This will enable understanding as to what is shown in the regional cell images. Following up on regional cell images, do these images snap the same subset of cells across the different architectures, as was done in Figure 4. If not, would this not make for a stronger comparison to showcase model performance on same cells across architectures.

Line above Equation 6: '... varying values of c '. What is c and where is c (or how is it encoded) in Equation 6? Either explain what is in quotes or cite the paper after 'penalty'.

Figure 5: (and related to comment 1.21): lower row says VAE expression. Is this now the ME-VAE expression? Similarly, the legend says 'individual VAE features'. Are these ME-VAE features?

Response 1.13: 'The size distribution can already be observed in Supplemental Figure 3 from the sampled cell images from varying places in UMAP space'. Supplemental Figure 3 does not match with author description.

Response 1.20: Supplemental figure 8 is not at all like Figure 5, as the authors claim.

Example of typos (this is just a partial list): devided, noticeable, what's, observe to has, downstream, Fig 5 legend, a maker of DNA expression....

Reviewer #2 (Remarks to the Author: Overall significance):

I would like to thank the authors for their efforts in addressing some of my previous comments. The manuscript is improved with additional experiments and clarifications.

However, some of previous comments remain unaddressed. The following are two examples of my previous comments that have not been addressed:

1. Response 2.4: Instead of arguing for the benefit of the VAE, I would suggest the authors conduct an experiment to support their claim, i.e, replacing VAE by a denoise auto-encoder, but with the same data argumentation/normalization. This would be a much cleaner way to justify the benefit of VAE.

2. Response 2.7: if the learned features cannot be generalized to new data, I would more likely question the utility/meaningfulness of the learned features from VAE. To me, it is essential to demonstrate the learned features are meaningful in a test dataset.

Reviewer #3 (Remarks to the Author: Overall significance):

The authors have responded to the concerns I raised previously. I only have minor comments regarding Fig 3.

Reviewer #3 (Remarks to the Author: Strength of the claims):

1. Fig3: The overall results should also include standard deviation in addition to the mean. I would

also suggest to include cluster purity and NMI separately for each cluster.

Minor issues in the text:

2. Page 9: "shows significant correlations shows correlates" -> "shows significant correlations"?
3. Page 13: "Rotation is corrected" -> "Rotation was corrected"
4. Page 18: "Clustermaps using hieracrchical clusters were calculated using the function's default method (Euclidean)." -> "Clustermaps using hierarchical clustering were calculated using the method's default distance metric (Euclidean)."
5. Github -> GitHub
6. CellPose -> Cellpose

Reviewer #3 (Remarks to the Author: Reproducibility):

The authors have shared the code, pre-trained models and a small test dataset. I was able to run the code with the test dataset after quick check through the code (self.data_dir + 'train/*' caused problems, does not work if self.data_dir does not end in '/'. Also might not work in Win. Better to use pathlib or os.path.join method.).

For future, might be useful for users if the authors would include one line example of calling the main.py script in README and also check the requirements (such as graphviz, Python libraries and their versions). Otherwise using the code was very straightforward.

Author rebuttal, second version:

Comments from Reviewer #1

Summary

The authors present ME-VAE, a DL architecture that removes known uninformative features by making them uniform and invariant across reconstructions, to improve downstream analysis of single cell imaging data. Using CYCIF images from MCF10A cells containing various biomarkers, the authors demonstrate that ME-VAE can clearly separate a TGF β +EGF population from controls, and outperforms standard VAE. The authors also show that there is a clear pattern of self-correlations between ME-VAE features, and identify representative clusters from these.

In the updated manuscript, the authors have incorporated changes to reflect most of the suggestions provided by the reviewers, especially the benchmarking to CODEX and other tools, and clarify comments for better understanding.

There still exists lack of clarity in the text and there are additional typos, wrong references introduced in the text which makes the rebuttal look quite unprofessional.

Specific Comments

Comment 1.1:

- In Response 1.5, authors explain that 'With the proposed methods it is necessary that the transformations of these features are known, but it is an unfortunate truth that if something cannot be calculated, it can't be corrected'. I would defer from this argument in that if something cannot be calculated, then attempts should be made to infer these so that they can be corrected. This is very true in biology as it is a mixed bag of many unknown and confounding features with very little that is known and the authors agree to this by stating the presence of 'countless artifacts and uninformative biological features'. If a model heavily relies on only the limited set of known features then this severely handicaps the model to generate new biological discoveries.

Response: 1.1:

- First, we would like to clarify that the statement “if something cannot be calculated, it can’t be corrected” does not mean we should not work to infer metrics for features that are not currently known. Take for example our shape correction method (Cells2Circles, which is described in the Methods). To our knowledge there was no known way to calculate and normalize shape, so we used registration to calculate a normalization for each cell. We encourage anyone who uses our method to experiment and infer new methods for quantifying and normalizing noise/features in images. By our statement, we are simply saying that for the ME-VAE to operate, a calculated transformation specific to each image must be applied as the transformation will not be applied by the architecture itself.
- Second, we would like to argue that current multiplex tissue imaging analyses heavily rely on only the limited set of known features such as mean intensity features computed across markers. On the other hand, the proposed method yields biologically meaningful representations (stain co-localization and subcellular pattern) by removing uninformative biological features, which often skew representation learning of VAE in undesired ways.
- Finally, as we mentioned in the previous rebuttal, potentially, if there are unknown noise-induced transformations, they will drive difference and skew the result so we could evaluate them and consider them as uninformative features by identifying transformations iteratively. For instance, rotation and polar orientation were first observed in Standard VAE so that we devise transformed images as a self-supervised signal to remove these uninformative features, which is our motivation and key contribution of the proposed work. These iterative processes will remove confounding features and refine biologically meaningful features, thus generating new biological discoveries.

Comment 1.2:

- Is Response 1.10 added to the main text? This will enable understanding as to what is shown in the regional cell images. Following up on regional cell images, do these images snap the same subset of cells across the different architectures, as was done in Figure 4. If not, would this not make for a stronger comparison to showcase model performance on same cells across architectures.

Response 1.2:

- The clarifications and reference made in previous response 1.10 have been added to the methods.
- Revised manuscript (Page 18): “Regional cell images within UMAP (Figure 2 and Supplemental Figure 4) were created by sampling cells from various regions of the UMAP embedding space to give visual context to the features that are being separated. Cells were placed into regions by snapping their embeddings to a grid and taking one representative image from each point on the grid as described by Schau *et al*²³.”

Comment 1.3:

- Line above Equation 6: ‘... varying values of c ’. What is c and where is c (or how is it encoded) in Equation 6? Either explain what is in quotes or cite the paper after ‘penalty’.

Response 1.3:

- Clarification for the c term and the quoted section have been added.
- Revised manuscript (Page 17): “In our application this means that the resulting latent space will ideally be independent of the undesired values injected into the architecture through the c vector:”

Comment 1.4:

- Figure 5: (and related to comment 1.21): lower row says VAE expression. Is this now the ME-VAE expression? Similarly, the legend says 'individual VAE features'. Are these ME-VAE features?

Response 1.4:

- We would like to thank the reviewer for pointing this out. Labeling this as ME-VAE expression would be more accurate and would help to avoid confusion with the VAE expression of the Standard VAE. We have updated the figures and legends accordingly.

Comment 1.5:

- Response 1.13: 'The size distribution can already be observed in Supplemental Figure 3 from the sampled cell images from varying places in UMAP space'. Supplemental Figure 3 does not match with author description.

Response 1.5:

- We apologize for the confusion. The size distributions are observed in Figure 3 and Supplemental Figure 4. No change to the manuscript is necessary to address this. All references to this figure in the text are correct.

Comment 1.6:

- Response 1.20: Supplemental figure 8 is not at all like Figure 5, as the authors claim.

Response 1.6:

- We apologize for the confusion. The added figure (which is highly visually similar to figure 5) is Supplemental Figure 7. No change to the manuscript is necessary to address this. All references to this figure in the text are correct.

Comment 1.7:

- Example of typos (this is just a partial list): devided, noticable, what's, observe to has, downstream, Fig 5 legend, a maker of DNA expression....

Response 1.7:

- We apologize for any typographical errors. These have been fixed in the text.

Comments from Reviewer #2

Summary

I would like to thank the authors for their efforts in addressing some of my previous comments. The manuscript is improved with additional experiments and clarifications.

However, some of previous comments remain unaddressed. The following are two examples of my previous comments that have not been addressed:

Specific Comments

Comment 2.1:

- Response 2.4: Instead of arguing for the benefit of the VAE, I would suggest the authors conduct an experiment to support their claim, i.e, replacing VAE by a denoise auto-encoder, but with the same data argumentation/normalization. This would be a much cleaner way to justify the benefit of VAE.

Response 2.1

- First, as requested by the reviewer, we implemented a denoising autoencoder using the same set of augmentations (rotation, polar orientation, size/shape – see Supplemental Figure 9). As we did with other architecture comparisons, we kept the encoder and decoder blocks the same and used the same latent dimensions for fair comparison. When we applied denoising autoencoder with the same set of augmented images, we observed that the denoising autoencoder is unable to separate the labeled perturbations (PBS and TGF β). While denoising autoencoders are very good at reconstructing corrupted images by removing noise, the lack of a regularization term on the latent space makes it less useful when being used to interpret embedded features. We also would like to argue that uninformative biological features described in the current manuscript are not simple noise or corrupted images. This interpretability of a continuous latent space is the main purpose of variational autoencoders and why they are used as the standard for embedding and extracting imaging features in the computer vision and biomedical imaging domain. Both of the methods that Reviewer 2 suggested for state-of-the-art comparison in the original revision utilize a variational distribution of latent spaces (Moyer et al for images and Romain Lopez et al for sc-transcriptomics).
- Second, our main purpose of this paper focuses on VAEs for single cell image analysis as many recent studies demonstrated that VAE approaches produced encouraging results by finding previously unappreciated cellular structures, allowing for accurate predictions across a variety of areas and outperformed classical methods for analyses of hand-crafted features. Thus, we do not believe this additional experiment to be pertinent to the main purpose of the paper and believe it would not be interesting to the majority of readers. We have therefore not added it to the main text but only included for rebuttal as the reviewer requested that we conduct an experiment to support our claim.

Comment 2.2:

- Response 2.7: if the learned features cannot be generalized to new data, I would more likely question the utility/meaningfulness of the learned features from VAE. To me, it is essential to demonstrate the learned features are meaningful in a test dataset.

Response 2.2:

- We would like to thank the reviewer for suggestion. As the reviewer requested, we have applied the trained model of the previous ME-VAE to another set of data (see figure below). Using a new replicate (unseen dataset) of PBS and TGF β +EGF, we segmented, normalized, and augmented the images following the methods described in the manuscript. The results are consistent with those shown in the main text, demonstrating the learned features are meaningful and further indicating good generalizability on top of the generalizability already illustrated with the additional multiplex imaging modality such as CODEX dataset. We do not believe this experiment to be pertinent to the main purpose of the paper and believe it would not be interesting to the majority of readers. We have therefore not added it to the main text.

Comments from Reviewer #3

Summary

The authors have responded to the concerns I raised previously. I only have minor comments regarding Fig 3.

Specific Comments

Comment 3.1:

- Fig3: The overall results should also include standard deviation in addition to the mean. I would also suggest to include cluster purity and NMI separately for each cluster.

Response 3.1:

- We have split the mean cluster purity to show individual cluster purities in a table format (Fig. 3). However, NMI is not an averaged metric the same way mean cluster purity is. NMI is used to compare collections of clusters/labels with a single metric, and it would not make sense to compute it separately for each cluster. The figure and caption have been updated to detail the results.

Comment 3.2:

- Page 9: "shows significant correlations shows correlates" -> "shows significant correlations"?
- Page 13: "Rotation is corrected" -> "Rotation was corrected"
- Page 18: "Clustermaps using hieracrchical clusters were calculated using the function's default method (Euclidean)." -> "Clustermaps using hierarchical clustering were calculated using the method's default distance metric (Euclidean)."
- Github -> GitHub
- CellPose -> Cellpose

Response 3.2:

- We apologize for any typographical errors. These have been fixed in the text.

Comment 3.3:

- The authors have shared the code, pre-trained models and a small test dataset. I was able to run the code with the test dataset after quick check through the code (self.data_dir + 'train/*' caused problems, does not work if self.data_dir does not end in '/'. Also might not work in Win. Better to use pathlib or os.path.join method.).
- For future, might be useful for users if the authors would include one line example of calling the main.py script in README and also check the requirements (such as graphviz, Python libraries and their versions). Otherwise using the code was very straightforward.

Response 3.3:

- We appreciate the reviewer's feedback regarding the usability of our code. GitHub code was updated with minor edits to make directories more straightforward for both windows, mac, and linux. We have also updated the README with an example of calling the main.py script.

Reviewer comments, Third version:

Reviewer #1 (Remarks to the Author: Overall significance):

I would like to thank the authors for their time and effort in addressing my previous comments. The manuscript reads well with the improved experiments and clarifications. I recommend acceptance.

Reviewer #2 (Remarks to the Author: Overall significance):

The authors have taken efforts to address my previous comments.

In the latest round of revisions, the authors presented results showing that:

- 1). denoising autoencoder (a vanilla version of VAE, their model) is substantially worse than VAE.
- 2). test results of their model on previously unseen data are as good as the results on the training data.

Based on our experience on VAE models, we usually would not expect to make such conclusions. For 1), I would expect denoising autoencoders might be slightly worse than VAE, but not substantially. For 2), I would expect the test results will not be as good. I feel the authors were either presenting partial results or having the experiments done incorrectly.

I have the following a few minor comments:

1. Put the two results, namely comparing with denoising autoencoder to justify the value of VAE, and generalization on previously unseen data to Suppl Material.
2. Instead of providing incomplete information as in the rebuttal letter, I recommend the authors to include full information, as in Figure 3a, for both points addressed in the response letter.
3. The authors should improve the writing of the Methods section. The description of ME-VAE is unclear and not presented with clarity. Notations in Equations 2,3,4 are simply confusing.

Author rebuttal, Third version:

Comments from Reviewer #1

I would like to thank the authors for their time and effort in addressing my previous comments. The manuscript reads well with the improved experiments and clarifications. I recommend acceptance.

Response: Thanks for your review.

Comments from Reviewer #2

The authors have taken efforts to address my previous comments.

In the latest round of revisions, the authors presented results showing that:

- 1). denoising autoencoder (a vanilla version of VAE, their model) is substantially worse than VAE.
- 2). test results of their model on previously unseen data are as good as the results on the training data.

Based on our experience on VAE models, we usually would not expect to make such conclusions. For 1), I would expect denoising autoencoders might be slightly worse than VAE, but not substantially. For 2), I would expect the test results will not be as good. I feel the authors were either presenting partial results or having the experiments done incorrectly.

Response: We would like to clarify these comments which we also discussed them with the editor separately.

1) our results actually show slightly worse performance for the standard VAE (0.53 vs 0.54 and 0.72 vs 0.66 – one cluster showed slightly better and the other cluster showed slightly worse). As we mentioned in the cover letter (previous revision), we showed that the Denoising Autoencoder performs similarly to the VAE despite being given additional information in the form of augmented images, which is somewhat expected result since Denoising Autoencoders do not have a regularized latent space and are primarily used for removing pixel level noise from reconstructed images, not creating a meaningful and interpretable latent space.

2) In fact, we reported the quantitative values of the test set are approximately 10% worse than the training (0.89 vs 0.80 and 0.88 vs 0.76). Reviewer 2's claims that the denoising autoencoder performed substantially worse than the standard and that our results on an unseen dataset are just as good as training are not consistent with the data and figures we produced for them.

Specific Comments

Comment 2.1:

- Put the two results, namely comparing with denoising autoencoder to justify the value of VAE, and generalization on previously unseen data to Suppl Material.

Response 2.1:

- The basic denoising autoencoder used with the same transformed images (as suggested by reviewer) have been included in Figure 2. This allows for the comparison of the standard variational autoencoder results with the denoising autoencoder results. Additionally, we have conducted an additional comparison between a multi-encoder using regularization in the loss function (ME-VAE) and a multi-encoder using just the reconstruction loss (ME-denoising AE). These results were added to the supplemental. The results of both sets of figures is consistent with prior findings. Both the VAE and DAE see a vast improvement when using multiple encoders for feature correction. Note that a key contribution of our paper is proposing "multi-encoder" architecture in conjunction with VAE (not VAE itself). To demonstrate generalizability or multi-encoder blocks, we tested ME-DAE and it showed improvement compared to the result of naïve DAE.
- The generalization results of the ME-VAE applied to an unseen replicate (previous comment from the Reviewer #2) have been added to the Supplemental Figures.

Comment 2.2:

- Instead of providing incomplete information as in the rebuttal letter, I recommend the authors to include full information, as in Figure 3a, for both points addressed in the response letter.

Response 2.2:

- The results include the full information with the same graphics and quantifications in Figure 2 as the other methods the DAE was compared to.

Comment 2.3:

- The authors should improve the writing of the Methods section. The description of ME-VAE is unclear and not presented with clarity. Notations in Equations 2,3,4 are simply confusing.

Response 2.3:

- This comment was raised in the first review, and thus we have addressed it in our first revision. As it did not raise again in the second revision, we thought that it was clearly addressed (also confirmed with other reviewers). We have made additional minor revisions to improve readability of the equations. Although we have made attempts to make the equations more interpretable, we do not know exactly what the reviewer finds confusing about them. They are fairly standard and commonly occurring equations within the field and the notation used here are similar to those used in other papers, including the paper reviewer 2 shared for the invariant C-VAE by Moyer et al. Moreover, the paper does not propose significant changes to the standard VAE loss functions. The only changes shown here are: 1) reflecting the transformed input in the loss function instead of the raw image and 2) averaging the KL-terms for the multiple encoders in the ME-VAE, both of which are described in the text. We would also like to add that the implementation of the ME-VAE is available on github (linked in the paper) with exact loss functions available using standard toolkits.

Comments from Reviewer #3

N/A